# Dissolved organic matter signatures in urban surface waters: spatio-temporal patterns and drivers

Clara Romero González-Quijano[1*], Sonia Herrero Ortega[2], Peter Casper[2], Mark O. Gessner[2,3], Gabriel A. Singer[1,4,5]

[1] Department of Ecohydrology and Biogeochemistry, Leibniz Institute of Freshwater Ecology and Inland Fisheries (IGB), Berlin, Germany

[2] Department of Experimental Limnology, Leibniz Institute of Freshwater Ecology and Inland Fisheries (IGB), Stechlin, Germany

[3] Department of Ecology, Berlin Institute of Technology (TU Berlin), Berlin, Germany

[4] Research Area of Earth, Hanse Wissenschaftskolleg, Delmenhorst, Germany

[5] Department of Ecology, University of Innsbruck, Innsbruck, Austria

*Correspondence to:* clara.romero@igb-berlin.de

## Abstract

Advances in analytical chemistry have facilitated the characterization of dissolved organic matter (DOM), which has improved understanding of DOM sources and transformations in surface waters. For urban waters, however, where DOM diversity is likely to be high, the interpretation of DOM signatures is hampered by a lack of information on the influence of land cover and anthropogenic factors such as nutrient enrichment and release of organic contaminants. Here we explored the spatiotemporal variation of DOM composition in contrasting urban water bodies, based on spectrophotometry and fluorometry, size-exclusion chromatography and ultrahigh-resolution mass spectrometry, to identify linkages between DOM signatures and potential drivers. The highly diverse DOM we observed distinguished lakes and ponds, which are characterized by a high proportion of autochthonous DOM, from rivers and streams where allochthonous DOM is more prevalent. Seasonal variation in DOM composition was apparent in all types of water bodies, apparently due to interactions between phenology and urban influences, such as nutrient supply, the percentage of green space surrounding to the water bodies and point source pollution. Optical DOM properties also revealed the influence of effluents from wastewater treatment plants, suggesting that simple optical measurements can be useful in water-quality assessment and monitoring, informing about processes both within water bodies and their catchments.

## 1 Introduction

Urban freshwaters typically receive high loads of organic carbon, nutrients and micropollutants, ranging from pharmaceuticals and personal care products to industrial chemicals and more (Schwarzenbach et al., 2006). Although routine wastewater treatment is increasingly effective, chemical stressors in urban freshwaters remain widespread. Prominent reasons are pollution legacies (Ladwig et al., 2017; Baume and Marcinek, 1993 ) and continued uncontrolled inputs, particularly by stormwater runoff (Council, 2009). In addition, urban surface waters tend to suffer from severe

hydromorphological modifications. This includes the lateral and vertical disconnection from floodplains and aquifers and results in large impacts on the extent and complexity of riverine habitat (White and Walsh, 2020). Furthermore, the disruption of connectivity limits the self-purification capacity of urban surface waters (D'arcy et al., 2007), which can lead to turbid water and visually unpleasant and potentially harmful algal blooms(Carpenter et al., 1998). This and the limited recognition of urban freshwaters as providers of ecosystem services (Huser et al., 2016) calls for improved

water management strategies that consider ecological in addition to hygienic and chemical criteria (Gessner et al., 2014).

The concentration and chemical composition of dissolved organic matter (DOM), generally quantified as dissolved organic carbon (DOC), are key characteristics of aquatic ecosystems. Both concentration and composition are governed by allochthonous inputs and internal biological production and transformation processes (Williams et al., 2016).

Typically, however, water quality monitoring only considers concentration and bulk quality properties (e.g. biological oxygen demand, BOD) as measures of DOM availability to, and degradation by, heterotrophic microbes (Jouanneau et al., 2014). This focus is at odds with the extreme diversity of DOM observed in freshwaters, where thousands of compounds can be chemically distinguished (Kellerman et al., 2014; Peter et al., 2020; Stanley et al., 2012). This high diversity and the strong spatio-temporal variation of DOM composition suggests much potential for DOM

characteristics to provide insights into the state of freshwater ecosystems in water quality assessment and monitoring. In fact, additional insights into freshwater ecosystems may be gained if the very high diversity of DOM can be used to inform about water quality for ecosystem assessment and monitoring purposes.

Recent progress in analytical methods has increasingly enabled the detailed characterization of DOM to elucidate the sources and fates in surface waters (Xenopoulos et al., 2021). Optical properties can inform not only about the chemical

characteristics of DOM but also, for example, about large-scale gradients in aquatic networks (Creed et al., 2015) or the degree of aquatic-terrestrial ecosystem coupling (Sankar et al., 2020; Lambert et al., 2015; Yamashita et al., 2010; Catalán et al., 2013). Fluorescence excitation-emission matrices (EEM) can be processed by parallel factor analysis (PARAFAC) to identify independently fluorescing DOM components (Cory and Mcknight, 2005). Size-exclusion chromatography partitions bulk DOM into molecular size fractions, which also tend to differ in origin and

bioavailability (Huber et al., 2011). Finally, the advent of ultrahigh-resolution mass spectrometry (FT-ICR-MS or Orbitrap-MS) has greatly refined the characterization of DOM, revealing associations between compositional turnover of DOM differing in molecular diversity and landscape-scale environmental gradients in lakes (Kellerman et al., 2014) and rivers (Peter et al., 2020).

In the present study we explored variation in the chemical composition of DOM over time and space in contrasting urban surface waters, hypothesizing that a detailed chemical characterization of DOM yields signatures of various human influences. To this end, we explored linkages between chemical composition of DOM and potential drivers determining DOM signatures, including land cover, eutrophication and chemical pollution, which we captured by using a suite of proxies. Our specific goals were to: (i) describe spatio-temporal patterns of DOM composition across a range of urban freshwaters encompassing streams, rivers, ponds and lakes; (ii) identify environmental factors accounting for the observed patterns; and thereby (iii) explore how information on DOM composition could be included in urban freshwater assessment and monitoring, complementing approaches and metrics currently used.

## 2 Methods

### 2.1 Study sites

The study was conducted in 32 freshwater sites located in the city of Berlin, Germany. Nearly 6.5% of the municipal area (889 km$^2$) is covered by freshwaters. These comprise 60 lakes (>1 ha), about 500 ponds, the two slow-flowing lowland rivers Spree and Havel, and numerous streams, ditches and canals. Selection of the 32 study sites followed a stratified random sampling design (Fig. 1a, Table A1). Based on geographical information for Berlin`s water bodies, we randomly selected 7 sites in each of 4 strata: lakes, ponds, rivers and streams. Rivers and streams were classified according to a width cutoff of 5 m. Monitoring data on water chemistry (Berlin city administration, SenUVK 2009-2014) were used in a cluster analysis to identify highly polluted sites. These were excluded from the pool used for randomly selecting study sites. Instead, two organically polluted rivers (H1 and H2) and streams (H3 and H4) were deliberately added to lengthen the environmental gradient. Sites H1 and H2 received WWTP effluents (Fig. 1a, Table A2) and sites H3 and H4 presented high levels of diffuse pollution. Other sites for, for some of which monitoring data were unavailable (streams and ponds), were also affected by pollution (Table A1): Pond P4 was formerly connected to an old waste water treatment plant and still receives stormwater inflow during heavy rain events; S5 is located immediately downstream of a WWTP; and R7 became a receiving stream in 2015 (Nega et al., 2019), which was too recent for the site to become classified as polluted based on the monitoring data. Land use data obtained from the Berlin city administration (Senate Administration for Environment, 2017) were used to calculate the proportion of paved areas and green spaces within a 50-m perimeter around the selected water bodies using open-source geoinformation software (Qgis Development Team, 2017). The 50-m perimeter was chosen to capture influences in the immediate vicinity of the sites, such as of the riparian zone and slightly beyond, but not of the whole catchments, which are highly variable in size and tend to be difficult to define in urban areas. Delineation of the 50-m perimeter enabled us to distinguish particularly between urban sites adjacent to paved surfaces vs. green spaces. Tufekcioglu (2020) and Johnson (2005) used buffer zones of similar size and a study on ponds using perimeters of up to 3200 m found 50 and 100 m to be most appropriate to assess land-cover effects (Declerck et al., 2006). All samples were taken during base flow conditions (Fig. E1).

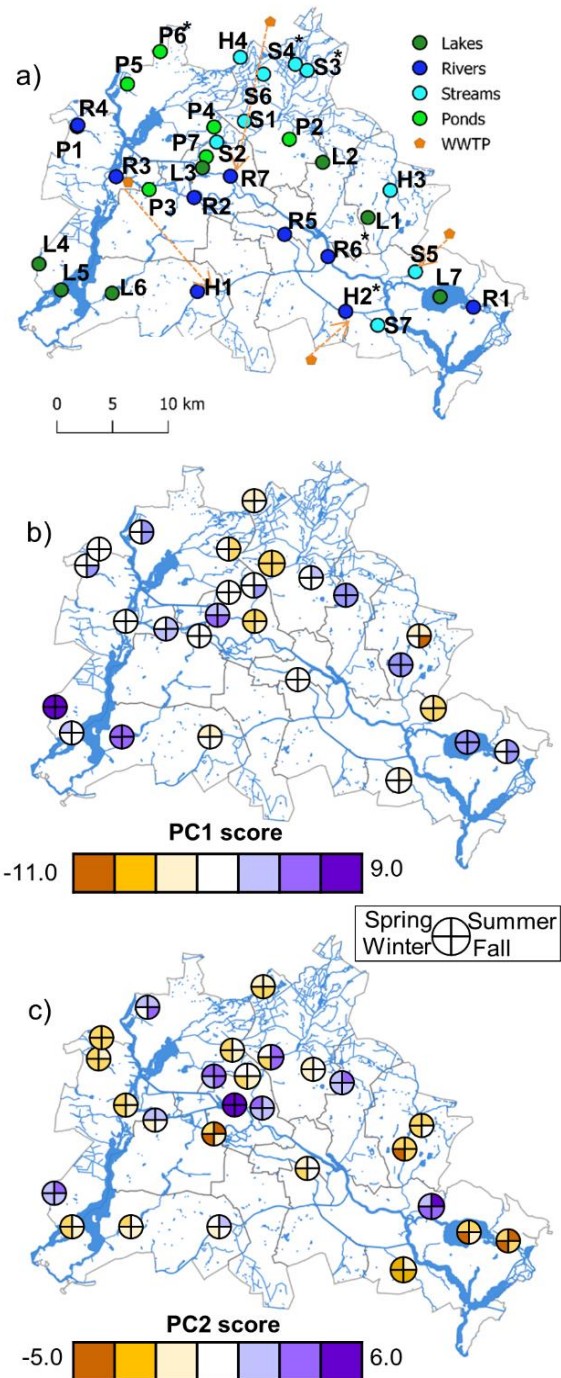

**Figure 1 Map of 32 sampling sites in the city of Berlin, including 7 lakes (dark green), 7 ponds (light green), 9 streams (light blue), and 9 rivers (dark blue), including two heavily polluted stream sites and two heavily polluted river sites. Wastewater Treatment Plants (WWTP) are shown in orange, arrows point to locations where the effluents are discharged (a). Scores of a Principal Component Analysis (PCA) of DOM characteristics are shown as color gradients for all sites sampled in four seasons (b,c). The PCA is based on DOC concentrations, all absorbance and fluorescence data, absolute component-specific fluorescence intensities from PARAFAC, and data from size-exclusion chromatography. Different colours indicate differences in DOM composition. Site codes are given in Table A1. Sites marked by asterisks (\*) were restricted to 3 seasons and hence excluded from the PCA.**

**2.2 Physico-chemical field measurements and water sampling**

We repeatedly sampled all 32 sites in each of four campaigns conducted over an annual cycle, first in spring (April-May 2016), then in summer (July-August 2016), autumn (September-October 2016) and winter (February-March 2017). All field visits occurred during base flow conditions (Fig. E1), We measured water temperature, pH, the dissolved oxygen (DO) concentration and electrical conductivity using a hand-held WTW Multiprobe 3320 (pH320, OxiCal-SL, Cond340i, Weilheim, Germany) or a smarTROLL probe (In-Situ, Fort Collins, CO, USA). We also

collected integrative water samples (2 L) from the upper 0.5 m water layer for chlorophyll-*a* and DOM analyses. The water was kept cool in acid-washed polycarbonate Nalgene bottles placed in a cooling box pending filtration in the laboratory (GF75, 0.3 μm average pore size; Advantec, Tokyo, Japan) within 6 hours after sampling. Additional volumes of surface water were filtered through pre-combusted glass fiber filters (GF75) directly in the field. These filters were placed in acid-washed, pre-combusted (450 °C, 4 h) glass vials (15-20 mL) sealed with a PTFE septum in

a screw-cap for later measurements of dissolved organic carbon (DOC) concentrations, DOM fluorescence and absorbance, and DOM molecular size distribution. The water passed through the filter was collected in acid-washed polyethylene tubes for analyses of soluble reactive phosphorus (SRP), nitrate ($NO_3^-$), nitrite ($NO_2^-$), ammonium ($NH_4^+$) and trace organic compounds (TrOCs). We also took unfiltered water samples for total phosphorus (TP) analysis. For each variable, we collected three replicate samples at each site in each season. We stored all samples in the dark in a

cooling box during transport. To preserve samples and remove all inorganic carbon, we acidified (pH 2) the water for DOC, $NO_3^-$, $NO_2^-$ and $NH_4^+$ analyses with 2 M HCl within 6 hours after sample collection. DOC concentrations and DOM fluorescence and absorbance were measured within 24 h. Filtered water for analyses of SRP, $NO_3^-$, $NO_2^-$, $NH_4^+$ and TrOCs was frozen at -20 °C.

**2.3 DOM characterization**

We determined total DOC concentrations by high-temperature catalytic combustion and infrared spectrometry on a TOC-V Analyzer (Shimadzu, Kyoto, Japan), with a 0.5 mg $L^{-1}$ limit of quantification and a typical analytical precision of 3%. DOM absorbance and fluorescence were simultaneously determined on an Aqualog instrument (Horiba Ltd, Kyoto, Japan) using ultra-pure water as a blank. From each site and season, we measured three analytical replicates of each of the three independent samples. We generated nine measurements (each of three process replicates was

measured 3 times) immediately after 3 blanks. The high level of replication allowed identification of artefactual measurements and outlier removal following a visual check of absorbance spectra and fluorescence excitation-emission matrices (EEMs). The fluorescence data was expressed in Raman units, removing the need for an external quantification standard.

         Iron can form stable complexes with DOC and interfere with optical DOC measurements, so that the two variables are
not independent (Maranger et al., 2003). We found that the quotient of light absorbance at 420 nm ($a_{420}$) and DOC concentration, a measure of the optical signal returned per unit DOC (Weyhenmeyer et al., 2014), was indeed significantly correlated (p = 0.002) with the Fe concentration measured in the monitoring program of the Senate of Berlin in two of our lakes (L5 and L7: $0.06 \pm 0.03$ mg/L), four of the rivers (H1, R1, R6 and R7: $0.30 \pm 0.14$) and two of the streams (H3 and H4: $0.31 \pm 0.14$). The relationship explained 31% of the overall variation (Fig. E2).

Consequently, Fe could have influenced our optical estimates of DOC concentration. However, because our analysis rests on differences in DOM composition as opposed to concentration (see below), it is unlikely that the presence of Fe notably influenced the spatial and temporal patterns observed in our study.

We calculated several indices from the absorbance spectra (Table B2): the specific UV absorption ($SUVA_{254}$) as a proxy for DOM aromaticity (Weishaar et al., 2003),  the ratio of absorbance at 250 and 365 nm (E2:E3) as an (inverse)

indicator of molecular size (Peuravuori and Pihlaja, 1997), the ratio of E4:E6 as an indicator of humification (Chen et al., 1977), and the ratio of slopes (SR) computed from short and long wavelength regions (Loiselle et al., 2009) as another negative correlate with DOM molecular weight. We used the fluorescence data to compute the freshness index $\beta/\alpha$ (Table B2) (Wilson and Xenopoulos, 2009), which indicates the relative importance of recently produced DOM (Parlanti et al., 2000). Furthermore, we calculated the fluorescence index (FI) as the ratio of fluorescence intensities at

the emission wavelengths of 470 and 520 nm (obtained at an excitation wavelength of 370 nm), which has proved useful to distinguish the relative contributions of terrestrial plants (FI~1.2) and microbes or algae (FI~1.4) as sources of DOM (Cory et al., 2010; Cory and Mcknight, 2005; Jaffé et al., 2008; Fellman et al., 2010). Finally, we computed the humification index (HIX) as a proxy for humic substances (Ohno, 2002). EEMs were used for PARAFAC, a multivariate three-way modeling approach decomposing EEMs into individual fluorophores (Bro, 1997; Stedmon and

Bro, 2008). We derived 7 components from a total of 116 EEMs and compared their loading spectra with the OpenChrom/OpenFluor database (http://www.openfluor.org) (Murphy et al., 2014). Prior to PARAFAC, we interpolated missing data in Rayleigh scatter regions to expedite the modeling process (Bro, 1997). The calculations for PARAFAC were performed using Matlab (version 7.11.0, MathWorks) and the DOMFluor Toolbox (1.7) following Stedmon & Bro (2008). We limited the number of components to 10, rigorously checked residual EEM plots, and

assessed the final models by split-half validation (Fig. B1) as recommended by Stedmon and Bro (2008).

The molecular size distribution of DOM was analyzed by liquid size-exclusion chromatography in combination with UV and IR detection of organic carbon and UV detection of organic nitrogen (LC-OCD-OND) (Huber et al., 2011). The instrument was calibrated with IHSS Suwannee River I Humic Acid and Fulvic Acid standards (International Humic Substance Society, St Paul, MN, USA). Carbon and nitrogen detectors were calibrated with potassium hydrogen

phthalate (C) and sodium nitrate (N). Limits of quantification were 0.1 mg C $L^{-1}$, and 0.01 mg N $L^{-1}$, analytical precision based on repeated standard measurements was better than 3%. We determined concentrations of three molecular size fractions: humic-like substances (HS-C and HS-N reported in mg C $L^{-1}$ and mg N $L^{-1}$, respectively), high-molecular weight non-humic substances (reported as HMWS-C and HMWS-N, in mg C $L^{-1}$ and mg N $L^{-1}$) and low-molecular weight substances (LMWS, in mg C $L^{-1}$).

To examine the molecular composition of DOM, we used ultrahigh-resolution Fourier-Transform Ion Cyclotron Mass Spectrometry (FT-ICR-MS). We extracted DOM on Agilent Bond Elut PPL solid-phase columns (Dittmar et al., 2008) from 1 L of filtered water acidified to pH 2. We then diluted extracts to 10 µg $L^{-1}$ C in 1/1 ultrapure water/methanol before broadband mass spectrometry on a 15 Tesla Solarix FT-ICR-MS (Bruker Daltonics, Bremen, Germany) in electrospray ionization negative mode (300 accumulated scans, ion accumulation time of 0.1 s, flow rate of 240 µL/h).

We performed internal mass calibration and exported the raw mass lists from 150 to 1000 Da for further data processing

using previously established R code (Del Campo et al., 2019). Briefly, we first applied a method detection limit similar to Riedel & Dittmar (2014) before aligning m/z values across samples (Del Campo et al., 2019). Subsequently, we assigned chemical formulas to mean m/z values assuming single-charged deprotonated molecular ions and Cl-adducts for a maximum elemental combination of $C_{100}H_{250}O_{80}N_4P_2S_2$, respecting chemical constraints. To eliminate doubtful formula assignments, we performed (i) an accurate assessment of mass error including its partitioning into random and systematic components (Savory et al., 2011): (ii) an exploration for stable isotope validation by daughter peaks (Koch et al., 2007), and (iii) a homologous series assessment based on $CH_2$, $CO_2$ and $H_2O$ as chemical building blocks for aliphatic, acid-based and alcohol-based elongation (Koch et al., 2007). To condense the mass-spectrometric data, we grouped formulas into 12 non-overlapping molecular groups (Lesaulnier et al., 2017) based on elemental composition and calculated the average molecular mass, number of formulas (molecular richness) and total intensity for each of them. In addition, we computed the double-bond equivalents (DBE) and the aromaticity index (AI) as indicators of unsaturated compounds, and the molecular lability boundary (MLB) as a measure of lability. Finally, we used van Krevelen plots to present the sum formulas derived from the FT-ICR-MS data in a space defined by O:C (oxygen richness) and H:C (saturation) ratios. We used random order of plotting to avoid bias due to systematic overplotting of thousands of compounds with identical O:C and H:C ratios.

**2.4 Additional water-chemical analyses**

$NO_3^-$, $NO_2^-$ and $NH_4^+$ were analyzed on a FIAcompact (MLE GmbH, Dresden, Germany). TP was measured using the same technique with unfiltered water samples that were digested with $K_2S_2O_8$ (30 min at 134 °C). We measured chlorophyll-*a* concentrations spectrophotometrically (HITACHI U2900, Tokyo, Japan) following hot ethanol extraction (Jespersen and Christoffersen, 1987) of three GF75 filters from each individual water sample. Concentrations of 18 trace organic compounds (TrOCs) were determined by HPLC-MS/MS (Shimadzu, Kyoto, Japan) (Zietzschmann et al., 2016). These included chemicals such as acesulfame (a sweetener), benzotriazole (a corrosion inhibitor), and drug residues like carbamazepine and gabapentin (Table C1).

**2.5 Data analysis**

We used repeated-measures ANOVA to test for differences among types of water bodies and sampling periods (referred to as seasons hereafter) for a variety of response variables; as the interaction between water body type and season was not significant we recomputed models including main effects only. Furthermore, we assessed the importance of seasonal variation in each water body type by computing a respective variance component using a type-II ANOVA (aka variance component analysis) for data from each water body type with season and site ID as random factors; this approach facilitates the assessment of temporal variation as a fraction of total variation within each water body type. Normal distribution was assessed graphically by quantile plots and histograms. For ANOVA, data were $\log(x)$ or $\sqrt{x}$ -transformed to achieve conditions of normality and variance homogeneity of the residuals.

For constrained multivariate analyses we considered land cover adjacent to the water bodies, trophic state and micropollutant load as drivers of variation in DOM chemical composition. We used the percentages of urban green space and paved areas as proxies for land cover, concentrations of TP, $NH_4^+$, $NO_3^-$ and chlorophyll *a* as a measure of

trophic state; and the mean TrOC concentration as a proxy for micropollutant load. We also performed a principal component analysis with all the TrOCs.

We followed a three-step approach to analyze the spatio-temporal patterns of DOM composition: First we identified major axes of variation in DOM composition by a PCA based on quantitative indicators of DOM, analytically accessible fractions thereof or quantitative proxies: DOC concentration, all absorbance and fluorescence indices, component-specific fluorescence intensities from PARAFAC normalized to DOC, and the size-exclusion chromatography data. Only the 27 sites sampled in all four seasons were included in this analysis. All variables were standardized to a mean of zero with a variance of 1 to ensure equal weighting, and projected onto the ordination space using Pearson correlations of the variables with PCA axes in a distance biplot (*sensu* Legendre and Legendre, 2012). To explore spatial patterns, we mapped PC1 and PC2 scores onto Berlin´s landscape using QGIS (QGIS Development Team, 2017).

Second, we used the same dataset as the dependent matrix in a redundancy analysis (RDA) with the set of potential drivers described above used as predictor variables. The goal of the RDA was to identify potential drivers of DOM composition and thereby assess, reciprocally, whether various DOM descriptors are ecologically informative. We started with the full RDA model and forward-selected drivers (Legendre and Legendre, 2012). For hypothesis tests in the RDA, permutations were restricted to account for repeated measurements at the same sites across seasons by first permuting sets of four seasonal measurements across sites and then permuting across seasons within each site. To check our ability to identify drivers behind major variation observed in DOM composition, we used Procrustes analysis to assess the similarity of PCA and RDA ordinations, including a permutation-based test of the non-randomness of the achieved superimposition (Mardia, 1979; Peres-Neto and Jackson, 2001).

Third, we exploited results of the FT-ICR-MS to facilitate interpretation of the two major axes of variation in DOM chemical composition resulting from the PCA. The FT-ICR-MS data were only available for three seasons and were purely compositional (relative intensities), as the many thousands of compounds contained in the spectra cannot be calibrated to yield concentrations. To link the quantitative and compositional datasets, we correlated PCA scores with compound-specific relative intensities of the mass spectra. The compound-specific correlation coefficients were then used as colour codes in van Krevelen plots. FT-ICR-MS-derived information such as the richness or average weight of specific molecular groups was also projected onto the PCA ordination space as arrows, provided correlation coefficients were >0.2. All statistical analyses and graphs were made with R 3.2.4 (R Core Team, 2016).

**3. Results**

**3.1 Physico-chemical characteristics**

Among all physico-chemical variables, only DOC concentration and temperature differed significantly among types of water bodies ($p < 0.05$ and $p < 0.001$, respectively). Temperature varied strongly across seasons, but still proved significantly different among water body types, with lakes and rivers being warmer than ponds and streams. DOC concentrations did not vary across seasons, but were significantly higher in ponds and streams than in lakes and rivers.

250 Ponds also showed the highest chlorophyll-*a* concentrations and rivers the lowest, but these differences were not significant.

Separate ANOVAs for each water body type showed that seasonal variation in TP and $NH_4^+$ concentrations was highest in rivers and streams (Table B1). Seasonal variation in $NO_3^-$ concentrations was generally high, but systematic differences were neither detected among seasons nor sites (Table B1). Seasonal variation of chlorophyll-*a* 255 concentrations was also high and similar across types of water bodies.

The analysis of TrOCs identified acesulfame, a widely used artificial sweetener (Buerge et al., 2009), in 72 out of a total of 120 samples taken at 32 sites across all seasons (Table C1). Similarly, two corrosion inhibitors, benzotriazole and methylbenzotriazole (Cotton and Scholes, 1967; Tamil Selvi et al., 2003), occurred in 68 and 63 samples, respectively. Fifteen other TrOCs were detected in at least 2 and up to 62 samples (Table C1). Rivers showed the 260 highest concentrations throughout the year. The first principal component of the PCA considering all TrOCs explained 61% of the total variance (Fig. C1) and separated streams and rivers with higher concentrations from ponds and lakes where concentrations of TrOCs were lower and often undetectable, particularly in ponds (Table C2). The strong positive correlations between most of the TrOCs suggested the applicability of a simple average TrOC concentration as a proxy for micropollutant load in further analysis; this mean was computed across all TrOCs after z-standardization 265 of each TrOC for equal weighting.

**3.2 DOM composition**

PARAFAC modeling resulted in 7 components referred to as C1-C7 (Table B3, Fig. B1). Components C6 and C7 were previously found to be protein-like, whereas all other components have been reported as humic-like (Table B3). In contrast to the standard physico-chemical variables and results from size-exclusion chromatography (Table B6), the 270 PARAFAC components and absorbance and fluorescence indices generally showed significant differences among water body types (Table B4 and B5).

The first axis of the PCA analyzing spatio-temporal patterns of DOM chemical composition explained 34% of the total variance (Fig. 2). PC1 was largely defined by the negative loadings for C1 and C2 (representing humic substances originating from wastewater treatment), $SUVA_{254}$ and LMWS (Fig. 2b). Furthermore, PC1 correlated positively with 275 the absorption slope ratio, E2:E3 (molecular size), β/α and HMWS-C. This axis separated water body types, from lakes on the right to ponds, rivers, and finally streams on the left. The optical proxies identified PC1 as a gradient spanning from lakes, where DOM had lower aromaticity and contained more freshly produced material, to streams, which showed high aromaticity and low proportions of fresh DOM. Pond P4, which was identified as an outlier because of particularly high $NH_4^+$ concentrations, also showed a rather distinct DOM composition.


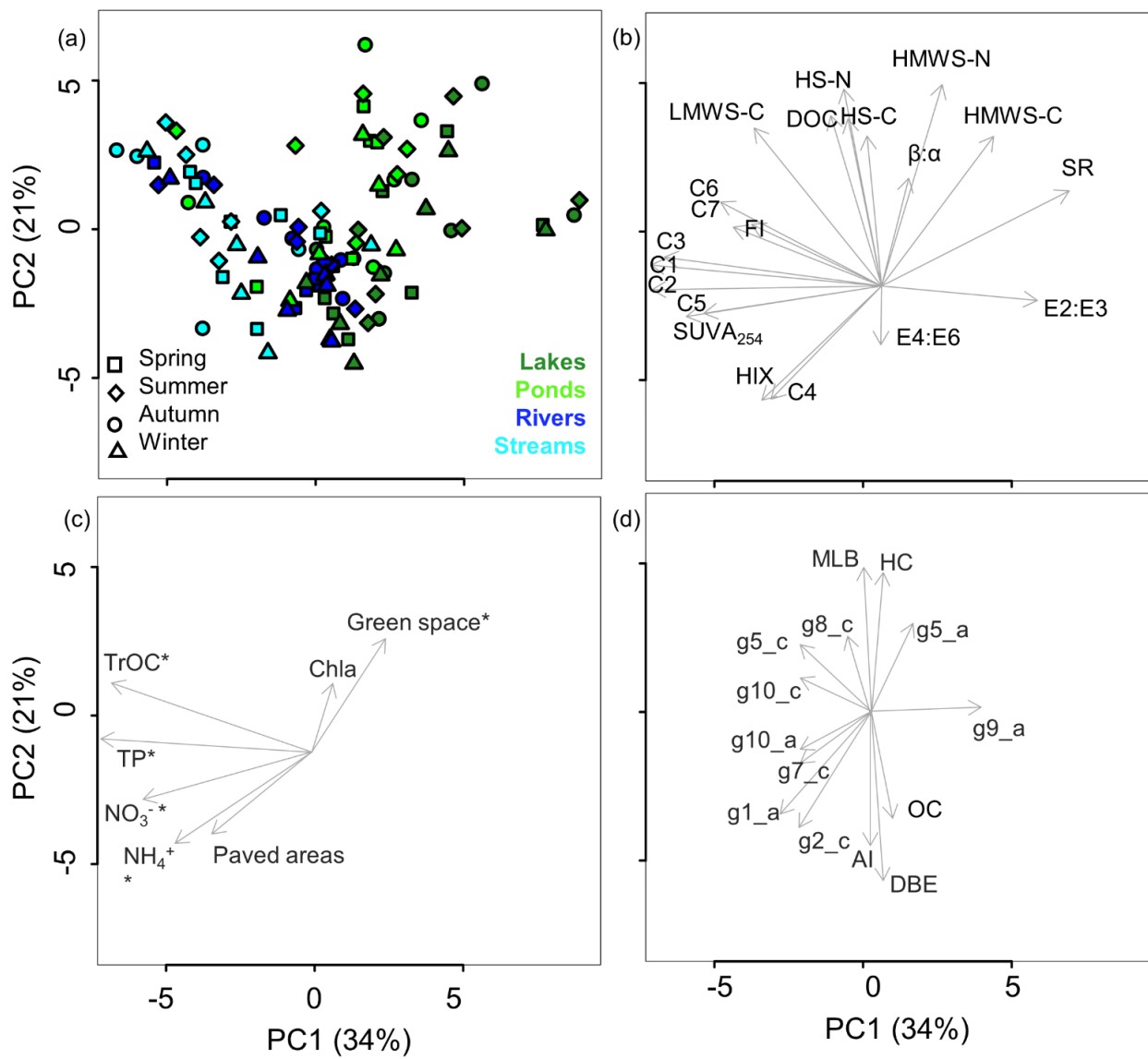

**Figure 2 Ordination of sites (a) by PCA based on DOM characteristics (b): (i) indices derived from measurements of absorbance (E2:E3 indicating molecular size, E4:E6 representing the humification ratio, the slope ratio SR, and SUVA₂₅₄) and fluorescence (freshness index β:α, fluorescence index FI, and humification index HIX), (ii) PARAFAC components C1**
**to C7, and (iii) data from size exclusion chromatography (humic-like substances HS, high-molecular weight non-humic substances HMWS, low-molecular weight substances LMWS). (c) Potential drivers of DOM composition, that were used as constraints in the RDA, were mapped onto the PCA ordination, with the significant constraints marked by an asterisk (*). (d) FT-ICR-MS-derived indices and molecular groups mapped onto the PCA ordination representing only groups correlated with PC1 or PC2 (r>0.2; oxygen richness O:C, saturation level indicated by H:C, double-bond equivalents DBE, aromaticity**
**index AI, molecular lability boundary MLB, molecular groups g1 and g2 indicating black carbon without and with heteroatoms, g5 consisting of unsaturated aliphatics, g7 representing saturated fatty acids, g8 and g9 denoting carbohydrates without and with heteroatoms N, S or P, and g10 comprising peptides). The molecular group measures are either average masses (marked by '_a') or counts of molecules (marked by '_c').**

PC2 explained an additional 21% of the total variance and correlated positively with HMWS (mg N/L) and β/α, and negatively with HIX. An exploration of spatio-temporal variation by plotting site-specific PC scores (Fig. 3) identified PC2 as the axis capturing temporal variation, with the four seasons aligning vertically at most sites. Winter and summer had the lowest and highest PC2 scores, respectively, with transitional seasons located in between. Thus, higher proportions of humic substances in winter contrast with more labile DOM in summer. In agreement with the variable-

specific seasonal variance components, the degree of seasonal differentiation differed among water body types also in multivariate space, being higher in streams and ponds than in the larger lakes and rivers (Fig. 3b). Except for site H3, seasonal variability was poorly reflected by PC1, which largely captured variation among individual water bodies or water body types, separating flowing from standing waters. Visual inspection of PCA scores mapped across Berlin (Fig. 1b,c) did not reveal a spatial signature transcending types of water bodies. RDA identified the areal percentage

of green space adjacent to the water bodies, TP, $NH_4^+$, $NO_3^-$ and the mean TrOC concentration as significant predictors of DOM composition (Fig. D1). The resulting PCA and RDA ordinations for DOM were strongly correlated (Procrustes rotation 0.73, p<0.001), suggesting that the considered predictors were indeed major drivers of variation in DOM chemical composition.

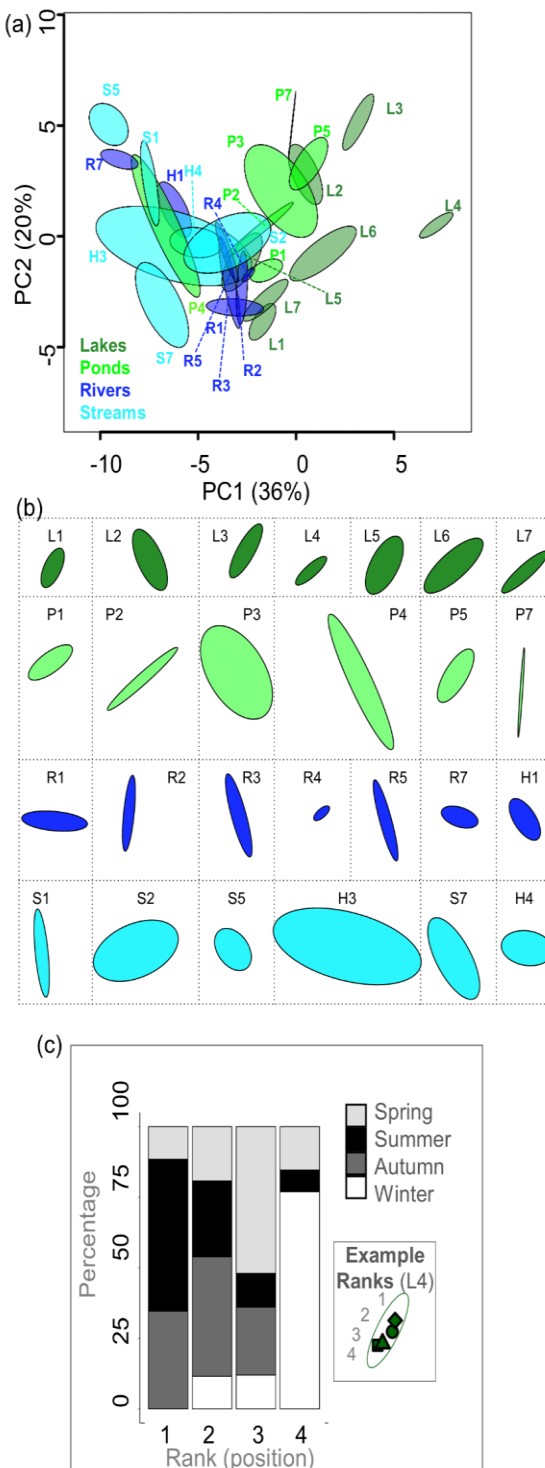

Figure 3 (a) PCA biplot based on DOM absorbance and fluorescence indices, PARAFAC components and size exclusion chromatography data from 4 contrasting types of urban freshwater bodies, including lakes, ponds, rivers, and streams, in addition to two streams and two rivers specifically selected as highly polluted sites. Each of the ellipses represents one sampling site that was visited 4 times, once in each season. Site codes are given in Table A1. (b) Visual comparison of site-specific seasonal variation based on the size, shape and orientation of ellipses, plotted separately per site. (c) Seasonal variation across sites illustrated by ranking the sampling dates at each site according to the PC2 scores, as shown in the

**inset. The stacked histograms show frequencies of the seasons across the four ranks. Summer samples tend to produce high scores at most sampling sites, whereas winter samples tend to score low.**

High-resolution mass spectrometric analyses of samples from three seasons provided additional insights into the chemical composition of DOM. Overall, we detected 6446 molecular formulas, most of them representing molecular

groups typical of humic material derived from soils. This includes highly unsaturated O-rich compounds, polyphenols and other aromatic structures, followed by unsaturated aliphatic polyphenols, and polycyclic aromatic compounds with aliphatic chains. The van Krevelen plots revealed a positive correlation of lignin-like molecules and carbohydrates with PC1 of DOM and identified these molecules as abundant in lakes (Fig. B2). In contrast, the negative association of proteins with PC1 was typical of streams. Information on the molecular groups identified by FT-ICR-MS and

projected on the PCA space (Fig. 2d) showed carbohydrates and sugars containing N, S or P to be positively related to PC1. Furthermore, PC1 was negatively related to black carbon, polyphenols and polycyclic aromatic compounds with aliphatic chains, which are all typical of soil-derived humic material, as well as with unsaturated aliphatics, saturated fatty acids and peptides, indicating that all of these molecular groups were more important in streams. Lastly, the computed molecular lability boundary (MLB), carbohydrates, sugars without heteroatoms (N, S or P) and unsaturated

aliphatics were positively related to PC2, while AI, DBE, black carbon and polyphenols were negatively related to PC2.

**4. Discussion**

**4.1 Spatial patterns and drivers of DOM signatures**

Our results show that the chemical composition of DOM in contrasting surface waters of the metropolitan area of

Berlin, Germany, is highly diverse. This reflects both aquatic-terrestrial linkages and DOM transformations within the aquatic systems (Fonvielle et al., 2021). Clear differences among the four types of water bodies we investigated were due to distinct signatures of streams and rivers vs. ponds and lakes. This was revealed especially by the first principal component (PC1) of a PCA (Fig. 2), which reflects the dominant gradient defined by variation in DOM composition across the 32 urban sites included in the study. Since optical measurements play an important role in our analysis of

DOM, it is important to consider potential interference by iron. Elevated iron concentrations lead to brownification, similar to effects of allochthonous DOM, and Fe and DOC also form stable complexes, so that the two variables are not independent (Maranger et al., 2003). However, Fe data available from the Senate of Berlin for two of our lakes (L5 and L7: $0.06 \pm 0.03$ mg/L), four of the rivers (H1, R1, R6 and R7: $0.30 \pm 0.14$) and two of the streams (H3 and H4: $0.31 \pm 0.14$), all concentrations were below the threshold of 1 mg/L, suggesting that Fe increases may result in

a420/DOC increases (Weyhenmeyer et al., 2014).

Stream DOM exhibited higher aromaticity (as indicated by $SUVA_{254}$) and lower amounts of recently produced, low-molecular DOM (as indicated by the freshness index or the slope ratio) than lakes at the opposite end of the gradient. This pattern matches results from agricultural streams near Berlin, where $SUVA_{254}$ values up to 3 L m$^{-1}$ mg$^{-1}$ were reported (Graeber et al., 2012) and from an urban river in southwestern Korea ($SUVA_{254}$ values of 2.5 L m$^{-1}$ mg$^{-1}$)

(Park, 2009). The distinct signature is also reflected in other DOM components, such as the fluorophore C2, which was more important in streams and identified as terrestrial humic material (Murphy et al., 2011). Streams also showed

higher levels of humic-like (C1) and protein-like (C7) compounds, whereas higher values of the freshness index characterized lakes. These patterns consistently indicate that the arrangement of sites along PC1 reflects a gradient of allochthonous vs autochthonous sources of DOM. A corollary of this finding is that despite the potentially pervasive influence of the urbanized surroundings, urban streams in particular are more tightly linked to the terrestrial environment than urban lakes, just as is the case for flowing and standing waters in natural landscapes (Larson et al., 2014).

In contrast to natural landscapes, however, the linkage of urban waters with their terrestrial surroundings is mediated by paved surfaces and engineered flow paths, including roof run-off into rain gutters, extensive (partially leaky) sanitation networks and sewage overflows in WWTPs that are activated following heavy rainfall or snowmelt. The urban gradient from allochthonous to autochthonous DOM sources we document could thus be driven by surface run-off rather than soil seepage and subsequent delivery of DOM to surface waters via groundwater Although we did not sample after major storms (Fig. E1), we would expect legacy effects of past runoff events to differ among sites, depending on the extent of green space and impervious surface area in the surroundings of the sites. This interpretation is supported by higher levels of proteins (Fig. 2) characterizing the urban streams and rivers, as opposed to soil-derived humic DOM signatures typical of unimpacted streams and rivers (Hutchins et al., 2017). The proteins could originate from surface runoff integrating various sources of urban pollution but they might also derive from WWTPs, as implied by the nature of some of the PARAFAC components (Table B3). For instance, the humic fluorophore C2 has been reported in WWTP effluents that may be discharged into urban surface waters (Murphy et al., 2011). Point-source inputs were also identified as drivers of DOM composition by the influence of TrOCs in our RDA and their correlations with C2 and C7, all of which are components of WWTP effluents.

Lakes differ from streams by a typically greater importance of autochthonous production. Since this production is fostered by abundant nutrient supply (given sufficient light), elevated nutrient concentrations should coincide with DOM signatures indicative of autochthonous carbon sources. This pattern has been found in agricultural streams, where the freshness index β:α indicating autotrophic activity was related to high nitrogen concentrations (Wilson and Xenopoulos, 2009). However, it contrasts with the negative relation between nitrogen concentration and the proportion of fresh DOM found across our study sites, where high nutrient concentrations were instead strongly related to DOM components of WWTP effluents. This typically resulted in an allochthonous DOM character at high-nitrogen sites. Notably, signatures like lower β/α in WWTP-impacted sites may also be a consequence of the highly processed nature of DOM that underwent degradation in a WWTP.

Similarly, the TP concentration was significantly related to DOM composition in our RDA, where phosphorus-rich water bodies also proved to have more allochthonous than autochthonous DOM. This points to inputs from urban surface runoff rather than groundwater inflow where long flow paths and residence times provide ample opportunities for phosphorus immobilization. As with N, additional phosphorus may derive from WWTP effluents, as suggested by the positive relationship between TP concentration and the fluorophore C2 as a putative tracer of WWTP effluents (Murphy et al., 2011). Overall, the negative relationships between nutrient availability and the importance of autochthonous components in the DOM pool suggests that while streams and rivers may efficiently collect N and P

from the urban environment; lakes are more efficient at channeling nutrients into autochthonous production. Thus, the autochthonous DOM signature in urban lakes appears to be largely independent of nutrient supply and rather be

facilitated by longer water residence times, higher water temperature and favorable light conditions.

Our results on urban surfaces driving urban allochthonous DOM composition meet our expectation that land cover notably influences the composition of DOM in urban surface waters  (Williams et al., 2016; Sankar et al., 2020). This conclusion is supported by results of our RDA, which identified the presence of green spaces in the perimeter of the water bodies as a significant influence. However, the relationship between land cover and DOM composition must be

interpreted with caution because all lakes were situated in areas with green spaces in their surroundings, whereas streams ran through areas dominated by buildings and paved surfaces. The urban running waters, more than lakes and ponds, thus received high surface runoff during rain events, including high inputs of pollutants and allochthonous DOM.

Except for ponds and some lakes, all investigated water bodies had direct surface water connections, which could result

in spatial autocorrelation. In addition, spatial patterns may arise from the prominent land cover gradients in Berlin, ranging from forested areas to densely populated urban centers. Since the sampling design of our study does not lend itself to a formal analysis of spatial autocorrelation, we explored spatial patterns with DOM proxies in maps (Fig. 1b,c) but found no obvious relationships. Instead, type-specific characteristics of the water bodies were pronounced, largely independent of hydrological connections. Factors potentially contributing to the resulting heterogeneity across the

surface waters in the city include specific local stressors such as point-source inputs of pollutants, spatially variable urban surface runoff delivering allochthonous DOM, and hydraulic-engineering structures such as sluices. Thus, our map of DOM composition (Fig. 1b,c) could be interpreted as visualizing heterogeneity in the conditions of urban surface freshwaters.

**4.2 Seasonal patterns and drivers of DOM signatures**

Seasonal variation in DOM signatures occurred in all types of water bodies mostly independent from variation among the four water body types. With a few exceptions, H3 being the most prominent example, seasonal variation of DOM composition was consistent across all water body types. (Fig. 3a,b), Assessed separately at each site (Fig. 3b), DOM was generally fresher in summer and autumn than in winter and spring, as indicated by higher ratios of β:α and more HMWS-N as indicators of polysaccharides and proteins (Thurman, 1985), whereas humic matter was more abundant

in winter, and the pattern in spring was not clear-cut. Our rank-based analysis of PC2 scores (Fig. 3c) suggests a consistent seasonal pattern of changes in DOM composition across sites, which emerged even though the variation within individual sites was limited along PC2.

At least four potential processes could account for the observed seasonal turnover in DOM composition: exudates of aquatic primary producers, microbial and sunlight-induced transformation of DOM, and terrestrial inputs from riparian

vegetation (Spencer et al., 2009; Cory et al., 2015), all of which could be influenced by the urban environment. Seasonal variation in light conditions could be important in influencing DOM composition by primary producers, independent of nutrient supply (see above), and temperature changes might also play a role, especially in determining

rates of microbial DOM transformations. Pulses of leaf litter falling or swept or blown into urban water bodies could be an additional source of DOM varying with season (Gessner et al., 1999). This holds particularly for urban green spaces and water courses lined by woody riparian vegetation. However, quantification of the relative importance of different drivers of seasonal patterns remains difficult based on the data currently available for urban settings.

The ponds and streams included in our study showed higher and less predictable seasonal changes in DOM composition than the lakes and rivers, as revealed by the pattern along PC2 (Fig. 3). This indicates that the nature and degree of aquatic-terrestrial coupling in urban settings leaves an imprint on seasonal changes in DOM composition. Therefore, the more extensive the time series data from surveys of DOM dynamics, the better can they inform about ecosystem conditions, complementing established procedures in water quality assessment and monitoring.

**4.3 DOM composition as a potential basis for urban surface water monitoring**

The fact that our analysis of DOM composition revealed specific characteristics of individual water bodies underlines the potential value of DOM descriptors as indicators that could be included in water-quality assessment and monitoring. Some sites deviated from the general pattern observed for water bodies of the same type. P4, for example, was formerly connected to an old waste water treatment plant and appeared to be influenced by previously unrecognized stormwater runoff. This legacy matches the particularly high levels of nutrients characterizing this site, especially $NH_4^+$, combined with a distinct DOM composition. Similarly, S5, located immediately downstream of a WWTP, although not specifically selected as a highly polluted site, also showed a distinct DOM composition as reflected by its highly negative PC1 score (Fig. 2a), indicating that the allochthonous influence was likely the strongest among all sites. Site R7 showed the same pattern as S5, and although not initially recognized as being affected by a WWTP, its DOM composition revealed that it had received WWTP effluents, which has actually happened since the end of 2015 (Nega et al., 2019). The distinct signatures at these individual sites are thus a promising starting point for incorporating information on DOM composition in water-quality assessment and monitoring. DOM optical indices would be highly cost-effective to apply and yield information that is not easily obtained by classic approaches. Robustness of such assessments would further increase when they are based on continuous time series. This could strengthen the implementation of current legal frameworks such as the EU Water Framework Directive aiming at an integrative water-quality assessment, including of urban water bodies.

**5. Conclusion**

The composition of DOM analyzed in a suite of contrasting water bodies of a large metropolitan area, the city of Berlin in Germany, is diverse, varying widely in molecular size and other features related to the degree of allochthonous inputs and conveying a distinct urban character. DOM features clearly differentiated water body types, from lakes with highly abundant autochthonous DOM to streams with more allochthonous DOM. Seasonal variation of DOM was prevalent in all water body types but likely to be driven not only by phenology but also by urban influences such as nutrient supply, WWTP effluents, reduced leaf litter input or flashy runoff resulting from sealed surfaces. Nutrient supply, the percentage of green space and concentrations of trace organic pollutants (as proxies for point source influences) were identified as drivers of DOM composition. In particular, simple optical measurements of DOM

characteristics were sufficient to detect WWTP effluents, a result that was corroborated by our data on TrOCs. This suggests that optical analysis of DOM could be a useful approach to complement current water-quality assessments

and monitoring. Such analyses are fast, inexpensive and easily implemented, and could be further supported by more sophisticated, potentially automated analyses such as the mass-spectrometric quantification of TrOCs. DOM composition can inform about processes both within water bodies and in the terrestrial surroundings; therefore, water-quality assessments could benefit from integrating information on DOM composition. Robustness of the approach would increase if the DOM assessments were based on time series or even continuous monitoring, for which

knowledge and technology are already available; this could indeed strengthen assessments as implemented in legal frameworks such as the EU Water Framework Directive.

**Appendix A includes a table showing site coordinates land cover and special features**

**Table A1: Coordinates, land cover, origin and special features. Longitude is given in decimal degrees East and latitude in decimal degrees North.**


| Site ID | Site name | Water body type | Latitude | Longitude | Agri-culture (%) | Forest (%) | Urban pave-ment (%) | Urban green space (%) | Origin |
|---------|-----------|-----------------|----------|-----------|------------------|------------|---------------------|----------------------|--------|
| H1 | Teltowkanal | River | 52.44239 | 13.32454 | 0 | 0 | 60 | 30 | Artificial |
| H2 | Teltowkanal | River | 52.42642 | 13.52039 | 0 | 0 | 100 | 0 | Artificial |
| H3 | Wuhle | Stream | 52.52562 | 13.57913 | 50 | 0 | 50 | 0 | Natural |
| H4 | Tegeler Fliess | Stream | 52.63442 | 13.38013 | 50 | 0 | 10 | 40 | Natural |
| L1 | Biesdorfer See | Lake | 52.50331 | 13.5497 | 0 | 0 | 50 | 50 | Artificial |
| L2 | Obersee | Lake | 52.54856 | 13.48972 | 0 | 0 | 50 | 50 | Artificial |
| L3 | Ploetzensee | Lake | 52.5438 | 13.33049 | 0 | 0 | 0 | 100 | Natural |
| L4 | Gross Glienicker | Lake | 52.46417 | 13.11489 | 0 | 10 | 0 | 90 | Natural |
| L5 | Havel | Lake | 52.4431 | 13.14453 | 0 | 0 | 100 | 0 | Natural |
| L6 | Schlachtensee | Lake | 52.44066 | 13.21183 | 0 | 60 | 30 | 10 | Natural |
| L7 | Müggelsee | Lake | 52.43837 | 13.6451 | 0 | 70 | 30 | 0 | Natural |
| P1 | Hoheheideteich | Pond | 52.57694 | 13.16428 | 0 | 100 | 0 | 0 | Natural |
| P2 | Hamburger Teich | Pond | 52.56738 | 13.44549 | 0 | 0 | 30 | 70 | Artificial |
| P3 | Ruhwaldteich | Pond | 52.52573 | 13.25998 | 0 | 0 | 50 | 50 | Artificial |
| P4 | Kienhorstbecken | Pond | 52.57724 | 13.34556 | 0 | 0 | 0 | 100 | Artificial |
| P5 | Mittelfeldteich | Pond | 52.61208 | 13.23045 | 0 | 100 | 0 | 0 | Artificial |
| P6 | Neurandteich | Pond | 52.63883 | 13.27377 | 0 | 0 | 65 | 35 | Artificial |
| P7 | Möwensee | Pond | 52.55282 | 13.33545 | 0 | 0 | 30 | 70 | Artificial |

| R1 | Müggelspree | River | 52.42985 | 13.68912 | 0 | 0 | 100 | 0 | Natural |
|---|---|---|---|---|---|---|---|---|---|
| R2 | Landwehrkanal | River | 52.51935 | 13.31959 | 0 | 0 | 80 | 20 | Artificial |
| R3 | Spree | River | 52.53613 | 13.21622 | 0 | 0 | 100 | 0 | Natural |
| R4 | Kuhlake | River | 52.57817 | 13.16509 | 0 | 100 | 0 | 0 | Natural |
| R5 | Neukölln Canal | River | 52.48936 | 13.43949 | 0 | 0 | 30 | 70 | Artificial |
| R6 | Spree | River | 52.47137 | 13.49683 | 0 | 0 | 100 | 0 | Natural |
| R7 | Panke | River | 52.5369 | 13.36759 | 0 | 0 | 60 | 40 | Natural |
| S1 | Zingergraben | Stream | 52.58209 | 13.38594 | 0 | 0 | 95 | 5 | Artificial |
| S2 | Schwarzer Graben | Stream | 52.56488 | 13.34918 | 0 | 0 | 50 | 50 | Natural |
| S3 | Graben 1 Buch | Stream | 52.62384 | 13.46883 | 0 | 100 | 0 | 0 | Artificial |
| S4 | Graben 73 Buchholz | Stream | 52.62881 | 13.45315 | 100 | 0 | 0 | 0 | Artificial |
| S5 | Erpe | Stream | 52.45888 | 13.61245 | 0 | 50 | 50 | 0 | Natural |
| S6 | Koppelgraben | Stream | 52.62065 | 13.41089 | 50 | 0 | 30 | 20 | Unknown |
| S7 | Plumpengraben | Stream | 52.41513 | 13.5628 | 0 | 0 | 100 | 0 | Natural |

**Appendix B includes tables that complement the physico-chemical and dissolved organic composition information**

Table B1: Physico-chemical characteristics (mean ± SD and % variance explained) of four contrasting types of water bodies in the city of Berlin. Means and standard deviations were computed across all seasons and sites. The percentages of variance explained (% Var) refer to the effect of season within each water body type, calculated by type-II ANOVA (aka variance component analysis), with season treated as a random factor. F-values refer to results of repeated-measures ANOVAs testing for differences among water body types (*** $p<0.001$, * $p<0.05$, ns = not significant).


| Water body type | Temperature (°C) | % Var | DOC (mg/L) | % Var | TP (mg/L) | % Var | $NH_4^+$ (mg/L) | % Var | $NO_3^-$ (mg/L) | % Var | Chlorophyll $a$ (µg/L) | % Var |
|---|---|---|---|---|---|---|---|---|---|---|---|---|
| Lakes | 14.6 ± 6.9 | 94 | 7.5 ± 2.7 | 23 | 0.05 ± 0.05 | 34 | 0.07 ± 0.07 | 26 | 0.22 ± 0.36 | 48 | 6.2 ± 14.2 | 61 |
| Ponds | 13.7 ± 5.3 | 94 | 10.3 ± 3.0 | 32 | 0.09 ± 0.07 | 30 | 0.27 ± 0.67 | 30 | 0.03 ± 0.06 | 52 | 7.3 ± 7.7 | 55 |
| Rivers | 15.2 ± 6.2 | 92 | 8.0 ± 1.6 | 13 | 0.10 ± 0.08 | 43 | 0.15 ± 0.14 | 68 | 1.12 ± 1.66 | 46 | 2.1 ±2.9 | 51 |
| Streams | 11.3 ± 4.9 | 87 | 11.7 ± 5.5 | 42 | 0.26 ± 0.31 | 21 | 0.36 ± 0.65 | 76 | 0.91 ± 1.53 | 43 | 5.3 ± 9.7 | 53 |
| $F_{water body}$ | 9.4*** | | 3.8* | | 2.8ns | | 1.3ns | | 2.5ns | | 1.0ns | |

**Table B2: Description of absorbance and fluorescence indices.**

| Variable | Description |
| --- | --- |
| SUVA$_{254}$ | Proxy for DOM aromaticity (Weishaar et al., 2003) |
| E2:E3 | Ratio of absorbance at 250 and 365 nm, as an (inverse) indicator of molecular size (Peuravuori and Pihlaja, 1997) (Chen et al., 1977) |
| E4:E6 | Indicator of humification (Chen et al., 1977) |
| SR | Ratio of slopes (SR) computed from short and long wavelength regions as another negative correlate with DOM molecular weight (Loiselle et al., 2009) |
| FI | Fluorescence index (FI) Ratio of the fluorescence intensities at the emissions 470 and 520 (obtained at excitation wavelength of 370nm). Indicator of DOM derived from terrestrial plants (FI around 1.2) or from microbes or algae (FI around 1.4) (Fellman et al., 2010; Cory and Mcknight, 2005; Cory et al., 2010; Jaffé et al., 2008) |
| HIX | Humification index (HIX) as a proxy for humic substances (Ohno, 2002) |
| β/α | Freshness index β/α (Wilson and Xenopoulos, 2009), which indicates the relative importance of recently produced DOM (Parlanti et al., 2000) |


**Table B3: Designation, excitation (Ex) and emission (Em) wavelengths of PARAFAC components, and the number of studies with matching components reported in OpenFluor (checked on the 28th March 2022) (Murphy et al., 2014).**

| PARAFAC component | Ex | Em | OpenFluor reference matches (0.95) | Explanation and selected references |
|---|---|---|---|---|
| C1 | 250 | 446 | 12 | Humic-like, peak A (Coble, 1996); humic-like and recalcitrant (C1) (Hansen et al., 2016) |
| C2 | 250 | 500 | 70 | Terrestial humic-like in waste water treatment impacted water, (G1) (Murphy et al., 2011); ubiquitous and recalcitrant humic (C2) (Chen et al., 2017) |
| C3 | 306 | 408 | 20 | Humic-like, peak M (Coble, 1996); humic-like (C3) (Stedmon and Markager, 2005) |
| C4 | 256 | 444 | 8 | Terrestrial humic-like, suggested as photo-refractory (C2) (Yamashita et al., 2010); terrestrial humic-like (C3) (Williams et al., 2013) |
| C5 | 250 | 382 | 12 | Anthropogenic, microbial humic-like (C6) (Williams et al., 2016) |
| C6 | 294 | 352 | 33 | Similar to tryptophan (C3) (Catalán et al., 2015); protein-like, linked to autochthonous production (C3) (Amaral et al., 2016) |
| C7 | 276 | 326 | 68 | Protein-like, peak B (Coble, 1996); waste water treatment protein (C2) (Teymouri, 2007) |


**Table B4: Variables of absorbance and fluorescence analyses (mean ± SD and % variance explained) in contrasting types of urban surface waters. Means and standard deviations were computed across all seasons and sites. The percentages of variance explained (% Var) refer to the effect of season within each water body type, calculated by a type-II ANOVA (aka variance component analysis), with season treated as a random factor. F-values refer to results of repeated-measures ANOVA testing for differences among water body types (\*\*\*p<0.001, \*\*p<0.01, \*p<0.05, ns = not significant). Abbreviations explained in Table B2**.

| Water body type | $SUVA_{254}$ | % Var | E2:E3 | % Var | E4:E6 | % Var | SR | % Var | FI | % Var | HIX | % Var | $\beta/\alpha$ | % Var |
|---|---|---|---|---|---|---|---|---|---|---|---|---|---|---|
| **Lakes** | 1.55 ± 0.39 | 20 | 8.99 ± 2.14 | 6 | 3.02± 1.34 | 67 | 1.38 ± 0.27 | 12 | 1.61 ± 0.08 | 69 | 0.77 ± 0.08 | 9 | 0.86 ± 0.09 | 12 |
| **Ponds** | 2.14 ± 0.51 | 37 | 6.65 ± 1.24 | 14 | 3.12± 0.74 | 46 | 1.20 ± 0.19 | 29 | 1.52 ± 0.06 | 61 | 0.83 ± 0.04 | 47 | 0.70 ± 0.04 | 35 |
| **Rivers** | 2.25 ± 0.15 | 55 | 7.04 ± 0.98 | 12 | 4.40± 14.27 | 80 | 0.017 ± 0.002 | 76 | 1.68 ±0.11 | 10 | 0.85 ± 0.03 | 24 | 0.79 ± 0.09 | 19 |
| **Streams** | 2.50 ±0.52 | 65 | 6.328 ± 0.876 | 54 | 3.53± 2.31 | 75 | 0.97 ± 0.13 | 22 | 1.63 ± 0.14 | 11 | 0.86 ± 0.05 | 21 | 0.73 ± 0.10 | 24 |
| $F_{water\ body}$ | 11.8\*\*\* | | 5.8\*\* | | 2.6 ns | | 9.2\*\*\* | | 3.5\* | | 4.9\*\* | | 5.5\* | |

**Table B5: PARAFAC components (mean ± SD and % variance explained) in contrasting types of urban surface waters. Means and standard deviations were computed across all seasons and sites. The percentages of variance explained (% Var) refer to the effect of season within each water body type, calculated by a type-II ANOVA (aka variance component analysis), with season treated as a random factor. F-values refer to results of repeated-measures ANOVA testing for differences among water body types (\*\*\*$p<0.001$, \*\*$p<0.01$, \*$p<0.05$, ns = not significant).**

| Water body type | C1 | % Var | C2 | % Var | C3 | % Var | C4 | % Var | C5 | % Var | C6 | % Var | C7 | % Var |
|---|---|---|---|---|---|---|---|---|---|---|---|---|---|---|
| **Lakes** | 0.17 ± 0.10 | 17 | 0.13 ± 0.06 | 14 | 0.24 ± 0.12 | 20 | 0.28 ± 0.12 | 14 | 0.31 ± 0.18 | 11 | 0.19 ± 0.10 | 15 | 0.18 ± 0.11 | 9 |
| **Ponds** | 0.24 ± 0.14 | 73 | 0.24 ± 0.11 | 80 | 0.40 ± 0.22 | 63 | 0.46 ± 0.21 | 63 | 0.52 ± 0.38 | 73 | 0.16 ± 0.11 | 65 | 0.25 ± 0.14 | 45 |
| **Rivers** | 0.51 ± 0.34 | 8 | 0.34 ± 0.17 | 12 | 0.66 ± 0.37 | 10 | 0.44 ± 0.12 | 32 | 0.61 ± 0.26 | 30 | 0.32 ± 0.22 | 14 | 0.25 ± 0.14 | 14 |
| **Streams** | 0.76 ± 0.47 | 16 | 0.57 ± 0.30 | 53 | 1.00 ± 0.58 | 30 | 0.85 ± 0.63 | 64 | 1.08 ± 0.73 | 16 | 0.37 ± 0.29 | 25 | 0.35 ± 0.18 | 45 |
| **Fwater body** | 6.3\*\* | | 10.4\*\*\* | | 7.6\*\*\* | | 7.2\*\*\* | | 8.4\*\*\* | | 2.2n.s | | 2.6n.s. | |

**Table B6: Results of size exclusion chromatography (mean ± SD and % variance explained) of samples from contrasting types of urban surface waters. Means and standard deviations were computed across all seasons and sites. The percentages of variance explained (% Var) refer to the effect of season within each water body type, calculated by a type-II ANOVA (aka variance component analysis), with season treated as a random factor. F-values refer to results of repeated-measures ANOVA testing for differences among water body types (\*\*\*p<0.001, \*\*p<0.01, \*p<0.05, ns = not significant). HS, humic-like substances; HMWS, high-molecular weight non-humic substances; and LMWS, low-molecular weight substances.**

| Water body type | HMSW | | HMSW | | HS | | HS | | LMWS | |
|---|---|---|---|---|---|---|---|---|---|---|
| | (mg C/L) | % Var | (mg N/L) | % Var | (mg C/L) | % Var | (mg N/L) | % Var | (mg C/L) | % Var |
| **Lakes** | $0.96 \pm 0.74$ | 27 | $0.11 \pm 0.07$ | 9 | $4.14 \pm 1.49$ | 19 | $0.25 \pm 0.11$ | 12 | $0.83 \pm 0.26$ | 18 |
| **Ponds** | $1.32 \pm 0.60$ | 41 | $0.16 \pm 0.06$ | 36 | $6.29 \pm 2.42$ | 26 | $0.33 \pm 0.13$ | 17 | $1.19 \pm 0.46$ | 45 |
| **Rivers** | $0.59 \pm 0.20$ | 60 | $0.09 \pm 0.03$ | 24 | $5.21 \pm 0.97$ | 33 | $0.31 \pm 0.09$ | 31 | $1.10 \pm 0.41$ | 22 |
| **Streams** | $0.73 \pm 0.45$ | 52 | $0.10 \pm 0.05$ | 41 | $7.21 \pm 3.75$ | 25 | $0.41 \pm 0.27$ | 9 | $1.48 \pm 0.62$ | 39 |
| $F_{water\ body}$ | 4.2\* | | 2.9ns | | 2.9ns | | 1.3ns. | | 3.7\* | |

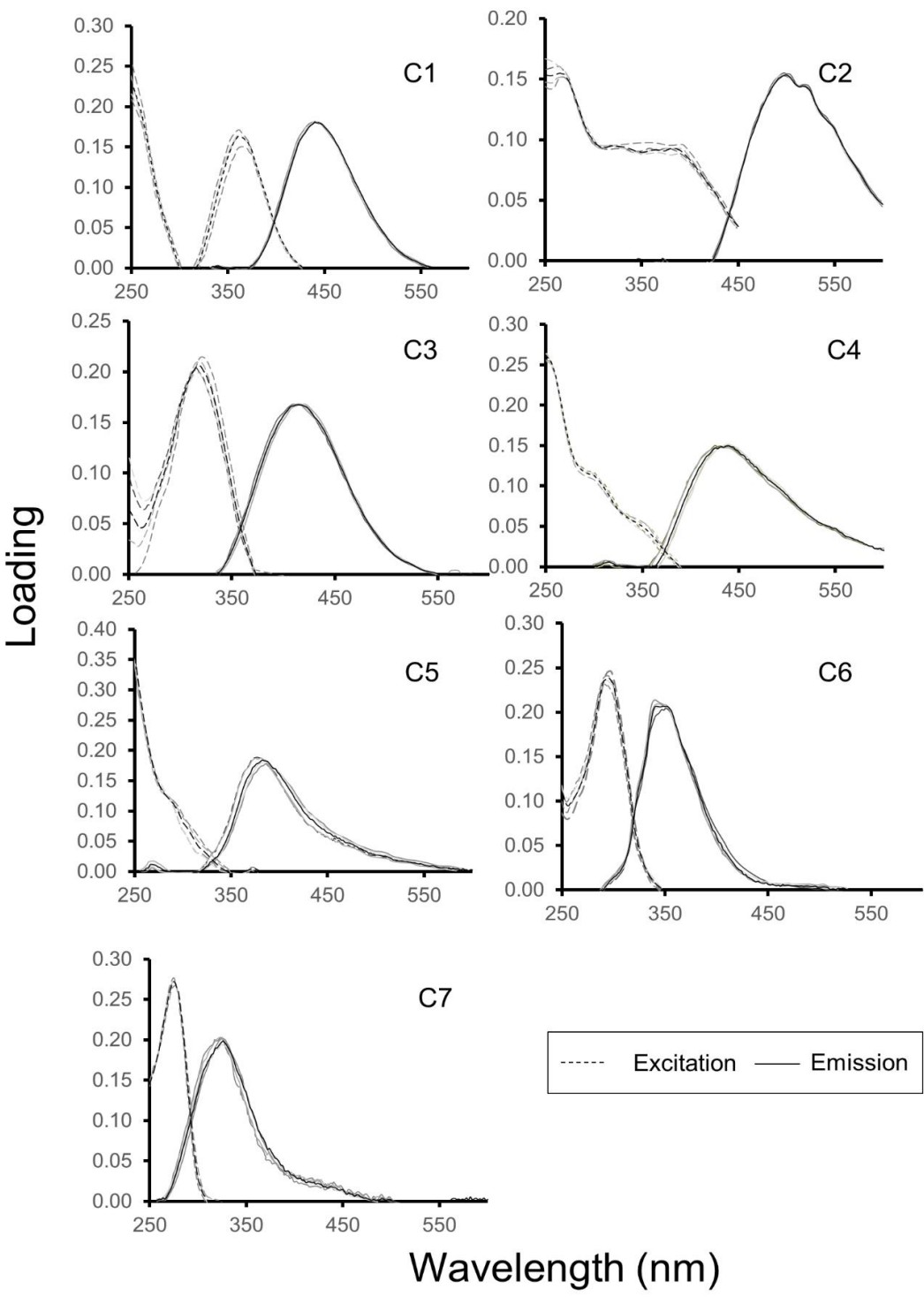

**Figure B1 Emission and excitation wavelengths of PARAFAC components. Solid lines represent emission spectra, dashed lines excitation spectra. Lines in different shades of grey refer to models using different sample sub-sets of a split-half validation analysis.**


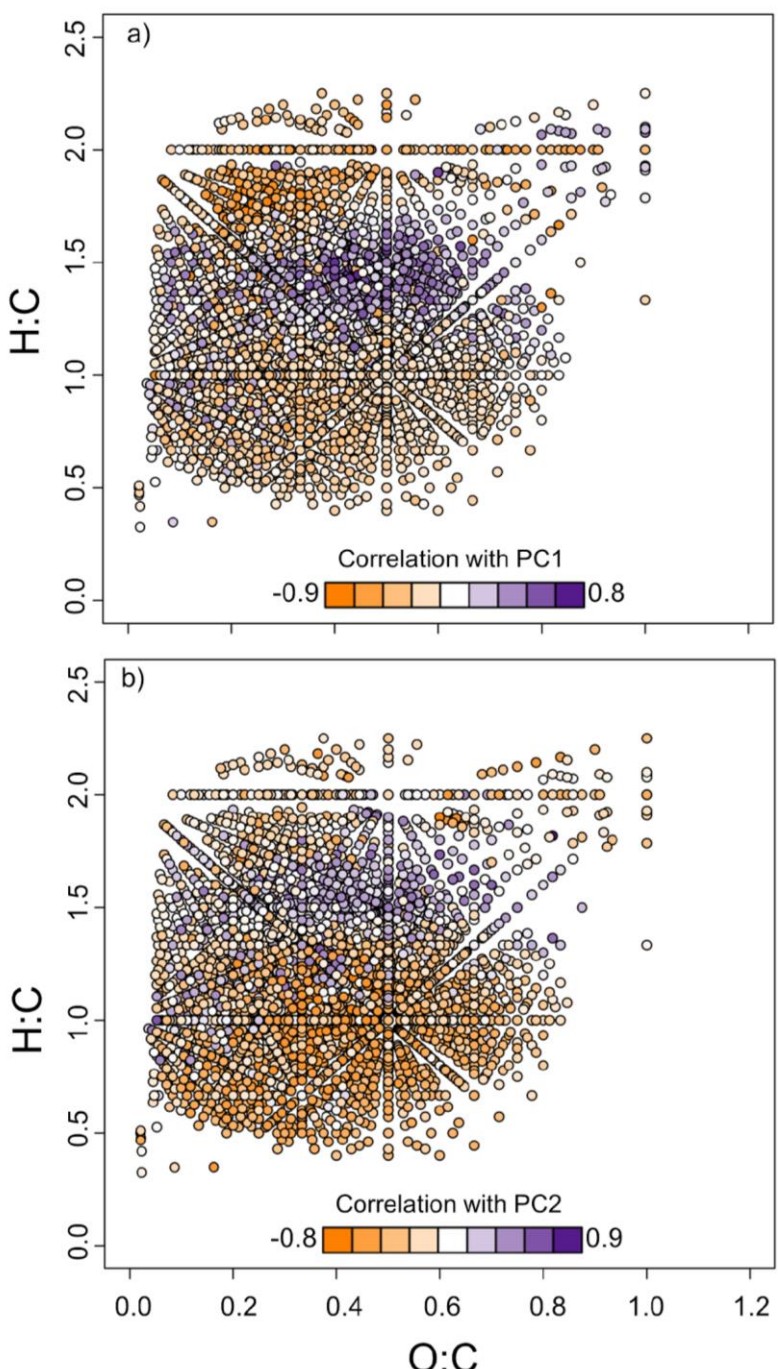

**Figure B2 Van Krevelen plots showing all molecules (sum formulas) identified by FT-ICR-MS analysis of DOM samples collected at 32 urban sites over three seasons (summer, autumn and winter). Colour indicates molecule-specific Spearman correlation coefficients of the relative intensities of each compound with the first (a) and second (b) axis of the PCA shown in Figures 2 and 3. The data points were plotted in random order to avoid bias resulting from identical O:C and H:C ratios for many sum formulas.**


**535** **Appendix C includes tables and figures that complement Trace Organic Compounds analysis**

Table C1: Trace organic compounds (TrOCs) analyzed in samples collected in urban surface waters. LLoQ = Limit of Quantification. Frequency refers to the number of occasions where concentrations exceeded the LLoQ.

| Acronym | LLoQ (µg/L) | Frequency | Name | Description |
|---------|-------------|-----------|------|-------------|
| ACS | 0.1 | 72 | Acesulfame | Sweetener |
| ATS | 0.05 | 42 | Amidrotrizoic | Radiocontrast agent |
| BTA | 0.1 | 68 | Benzotriazole | Corrosion inhibitor |
| BZF | 0.1 | 6 | Benzafibrate | Lipid-lowering agent |
| CBZ | 0.05 | 44 | Carbamazepine | Anticonvulsant |
| DCF | 0.05 | 30 | Diclofenac | Analgesic/anti-inflammatory agent |
| FAA | 0.1 | 46 | 4-formylamin metabolite of metamizol | Analgesic |
| GAB | 0.1 | 62 | Gabapentin | Drug for epilepsy treatment/pain killer |
| GPL | 0.05 | 39 | Gabapentin-lactam | Derivate of gabapentin |
| IOM | 0.1 | 40 | Iomeprol | Radiocontrast agent |
| IOP | 0.01 | 52 | Iopromide | Radiocontrast agent |
| MBT | 0.1 | 63 | Methylbenzotriazole | Corrosion inhibitor |
| MTP | 0.1 | 31 | Metoprolol | Beta blocker |
| PRI | 0.05 | 31 | Primidone | Anticonvulsant |
| SMX | 0.1 | 2 | Sulfamethoxazole | Antibiotic |
| VAL | 0.1 | 30 | Valsartan | At1-receptor antagonist |
| VLX | 0.1 | 3 | Venlafaxine | Antidepressant |
| VSA | 0.1 | 62 | Valsartan acid | Antihypertensive agent |

**Table C2: Mean concentrations and standard deviations of Trace Organic Compound (TrOC) per water body type. See Table C1 for full names. BZF, SMX and VLX were always below the limit of quantification (LLoQ) and are hence omitted from the table.**

| Acronym | TrOC concentration (µg/L) | | | |
|---------|-------|-------|--------|---------|
|         | **Lakes** | **Ponds** | **Rivers** | **Streams** |
| ACS | $0.23 \pm 0.17$ | $0.15 \pm 0.16$ | $0.28 \pm 0.17$ | $0.78 \pm 1.35$ |
| ATS | $0.08 \pm 0.12$ | <LLoQ | $0.74 \pm 1.12$ | $0.43 \pm 0.88$ |
| BTA | $0.34 \pm 0.51$ | $0.38 \pm 0.91$ | $2.37 \pm 3.31$ | $2.16 \pm 3.61$ |
| CBZ | $0.07 \pm 0.08$ | <LLoQ | $0.37 \pm 0.48$ | $0.41 \pm 0.66$ |
| DCF | <LLoQ | <LLoQ | $0.97 \pm 1.38$ | $0.88 \pm 2.14$ |
| FAA | $0.15 \pm 0.23$ | <LLoQ | $1.25 \pm 1.66$ | $2.10 \pm 4.25$ |
| GAB | $0.27 \pm 0.36$ | <LLoQ | $0.42 \pm 0.43$ | $0.74 \pm 1.12$ |
| GPL | $0.10 \pm 0.17$ | <LLoQ | $0.19 \pm 0.42$ | $0.13 \pm 0.18$ |
| IOM | $0.18 \pm 0.28$ | <LLoQ | $1.18 \pm 2.36$ | $1.44 \pm 2.91$ |
| IOP | $0.09 \pm 0.19$ | $0.01 \pm 0.02$ | $0.27 \pm 0.32$ | $1.46 \pm 3.79$ |
| MBT | $0.27 \pm 0.39$ | $0.11 \pm 0.24$ | $0.91 \pm 1.01$ | $0.69 \pm 1.27$ |
| MTP | <LLoQ | <LLoQ | $0.47 \pm 0.58$ | $0.63 \pm 1.55$ |
| PRI | $0.03 \pm 0.02$ | <LLoQ | $0.16 \pm 0.22$ | $0.26 \pm 0.52$ |
| VAL | <LLoQ | <LLoQ | $0.39 \pm 0.44$ | $0.97 \pm 3.65$ |
| VSA | $0.70 \pm 1.02$ | <LLoQ | $3.22 \pm 3.84$ | $3.33 \pm 5.72$ |

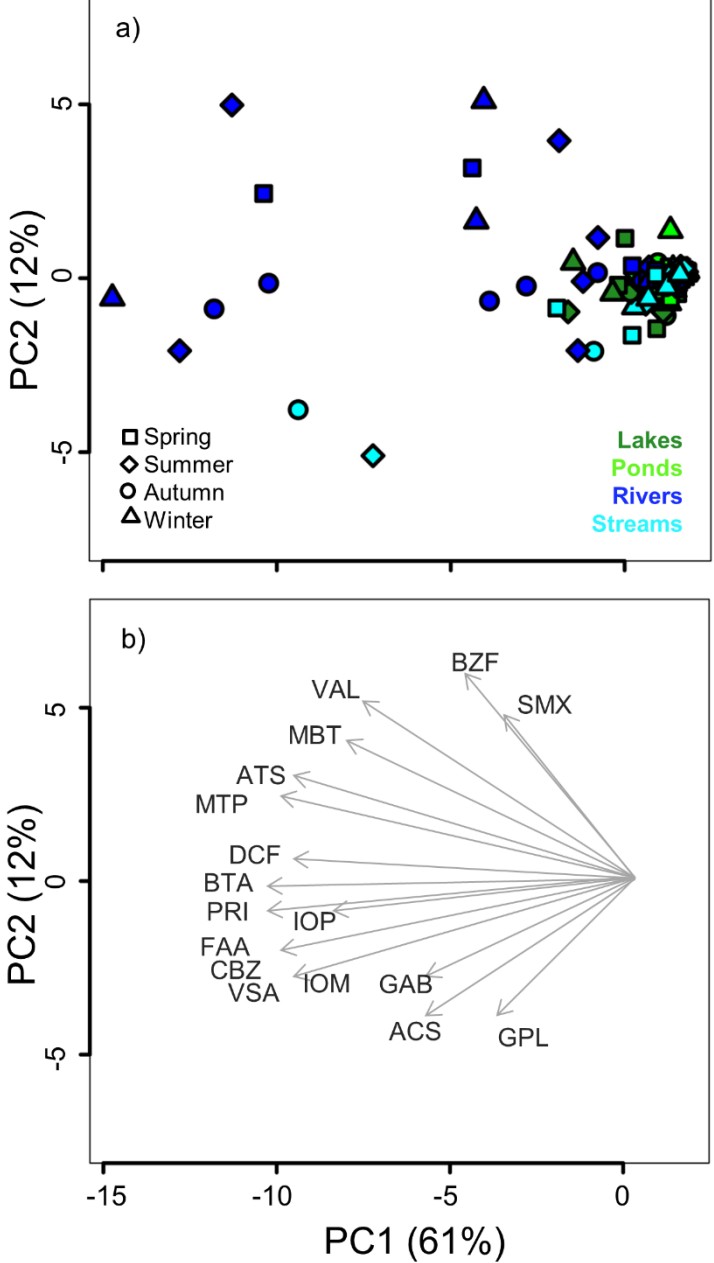

**Figure C1 Principal Component Analysis (PCA) of 32 urban sites in the city of Berlin over four seasons (a) and Trace Organic Compounds (TrOCs) (b). Site S5 had extreme PC1 and PC2 scores; the site was included in the analysis but is not presented in the biplot to better visualize variability among the other sites. Abbreviations of the TrOCs (B) are explained in Table C1.**

**Appendix D includes the RDA analysis**

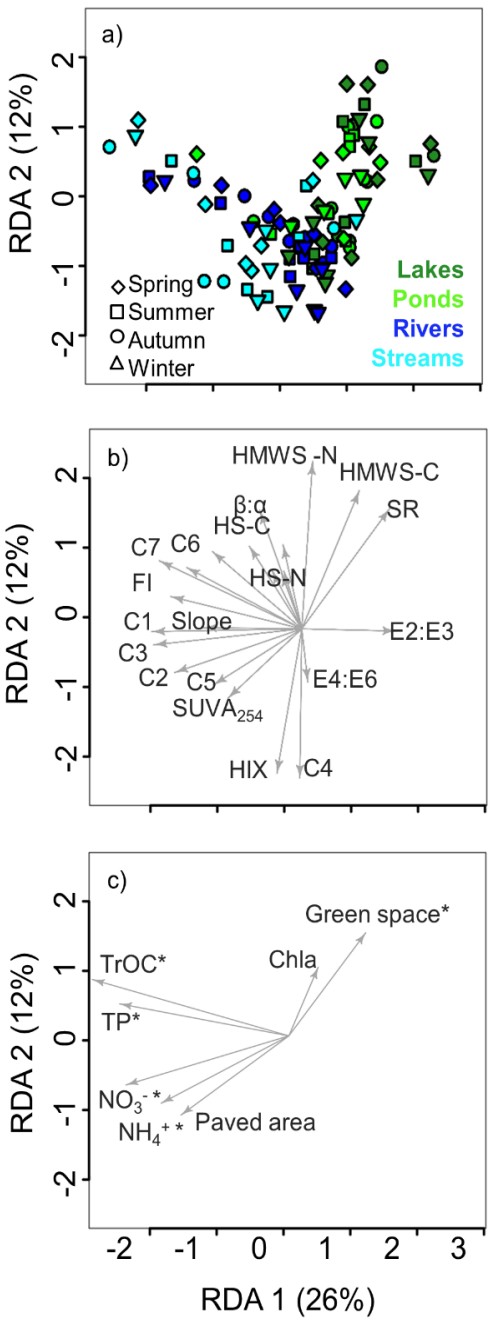


**Figure D1 Redundancy Analysis (RDA) of urban sampling sites (a) visited 4 times over one year, the DOM characteristics included in the analysis (b) and the predictor variables (c), the last marked by an asterisk (\*) when significant. DOM characteristics include (i) absorbance and fluorescence indexes (E2:E3, molecular size, E4:E6, indicator of humification, SR, slope ratio, β:α, freshness index, SUVA$_{254}$ and HIX, humification index), (ii) PARAFAC components (C1 to C7), and**

**(iii) fractions derived from size exclusion chromatography (HS, humic-like substances; HMWS, high-molecular weight non-humic substances; and LMWS, low-molecular weight substances).**

**Appendix E Includes precipitation and flow in the city of Berlin during the study and the relation of Iron and absorbance at 420 relative to DOC.**

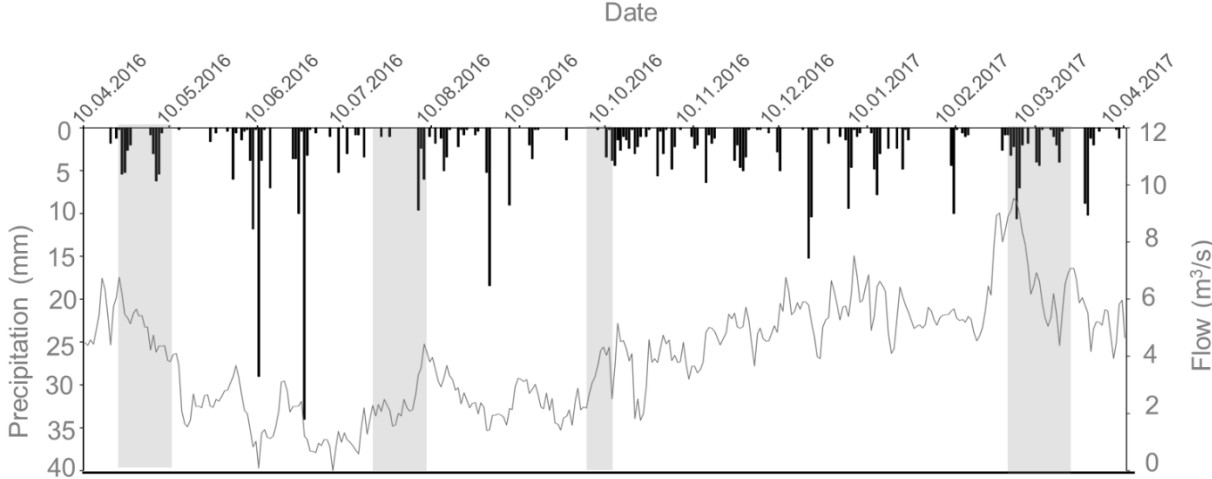


**Figure E1 Precipitation and flow at a site within the city of Berlin during the study period, with the grey boxes indicating the four sampling periods.**

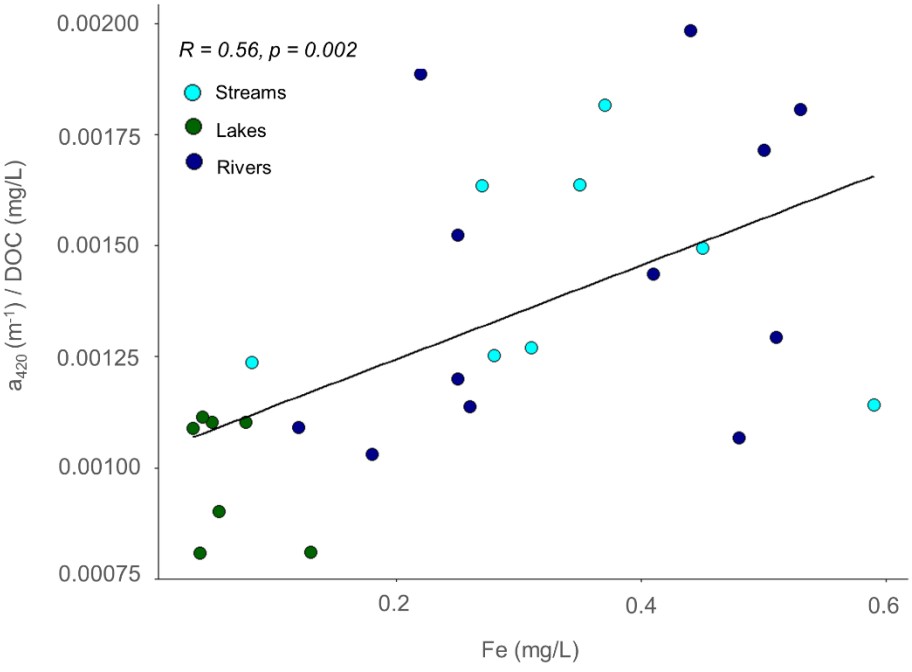

**Figure E2 Relationship between iron (Fe) and the absorbance at 420nm relative to DOC(a420/DOC)**

*Author contributions*. All authors contributed to designing the study. CR and SH collected the data. CR did the optical analysis and the PARAFAC modeling, GS carried out the FT-ICR-MS analysis. CR and GS conducted the statistical analysis. CR led the manuscript writing, jointly with GS. All authors discussed results and edited the manuscript.

*Competing interests*. The authors declare not to have a conflict of interest.

*Data availability*. The data will be made available at ttps://www.pangaea.de.

*Acknowledgments*. We thank A. Köhler at the Senate Berlin (SenUVK) for water quality data, authorities and private land owners for providing access to the study sites, C.N. Stratmann for obtaining permissions, U. Mallok for nutrient analyses, C. Schmalsch for the LCOCD analysis, S. Krocker and T. Fuss for the DOC analysis and T. Goldhammer for advice with chemical analyses. C.N. Stratmann, Meinhold, I. Ajamil, G. Idoate, L. Thuile-Bistarelli, A. Sultan, R. Schulte, E. Tupper, T. Fuss, R. del Campo, A. Wieland, and M. Bethke for field assistance. G. Aschermann and A. 575 Putschew kindly enabled TrOC analyses. Access to FT-ICR-MS and associated expertise was generously provided by T. Dittmar during a stay of G. Singer at the University of Oldenburg that was funded by the Hanse-Wissenschaftskolleg Delmenhorst. Thank you also to K. Pypkins for support with GIS and to B. Kleinschmit for thoughts on the sampling strategy and data analysis. This project was funded by the German Research Foundation (DFG) through the Research Training Group 'Urban Water Interfaces' (UWI; GRK 2032).

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
