# Peer review of "Dissolved organic matter signatures in urban surface waters: spatio-temporal patterns and drivers"

_Biogeosciences, 2021_

## Author Response (AR1)

Dear Professor Ji-Hyung Park,

Thank you for the constructive comments, which we have addressed in a revised version of the manuscript as detailed below.

**COMMENTS FROM THE EDITOR**

*-* **More details about QA/QC: In addition to the blank analysis commented by a reviewer, please provide more details such as analytical accuracy, reference material and any other QC measures for DOC analysis and other DOM characterization techniques.**

*We have added additional details about QA/QC with regard to DOC (L143-144), optical analyses (L145-150) and molecular size distribution (L 185-188).*

*-* **Given the large variations in measured values, some data may not have conformed to the assumption of normal distribution. It would be helpful if you articulate in section 2.5 as to how you checked up the normality and, if it was the case, handled data exhibiting the non-normal distribution.**

*Normal distribution was assessed graphically by quantile plots and histograms. For ANOVA, data were log(x) or √x -transformed to achieve conditions of normality and variance homogeneity. We provide this information now in L237-239.*

*-* **You wanted to include rainfall data to respond to a reviewer comment on hydrological impacts. Given the differential patterns of DOC concentrations and DOM optical properties between the rising and descending limbs of the hydrograph, I would suggest that you show the sampling times on the flow curve as part of a multi-plot including precipitation and flow. You could take flow data from a representative gaging station adjacent to your sampling sites.**

*We have included such a figure in the Appendix (Fig. E1) showing precipitation and river flow. Both are overlain with the sampling occasions for ease of reference.*

*-* **Sorry, but I have to draw your attention to careful language editing. A careful editorial check-up during the revision would be required to remove typos such as spectrophometry (L 19) and clarify some vague sentences like "The highly diverse DOM we observed distinguished lakes and ponds characterized by…" (L 20-22) throughout the manuscript.**

*We have corrected the typos we detected and rephrased all sentences that appeared not to be clear enough.*

**I would like to ask you to make all the changes easily identifiable in a marked-up manuscript based on your point-by-point responses to the reviewer comments. If possible, please specify the line numbers of the revised parts in your responses accompanying the revised manuscript.**

*As suggested, we have included line numbers (from the document with tracked-changes) in our point-by-point responses to the reviewers' comments.*

**ANSWER REVIEWER 1**

**Overall, this is an interesting dataset and questions.**

*Thank you*

**The results and discussion are somewhat challenging. The discussion and results lack a clear organizing structure.**

*We have improved the links between the Methods, Results and Discussion sections to better structure the manuscript. We realized that information about some sites was first mentioned in the Discussion only, which made it difficult to follow. Therefore, we have improved the description of the sites in Methods. This concerns particularly information on WWTP effluents and other specific features of various sites. We also now introduce the idea of DOM monitoring upfront in the Introduction to return to it in the last section of the Discussion.*

**Some of the relationships claimed in the analysis have no clear/singular interpretation. For example, the authors say that C2 and C8 are associated with wastewater because authors have said they were associated with wastewater in other study in other regions of the world and because they are associated with TrOCs. A similar statement is made about nutrients. The problem, however, is that all of the components ordinate in the same direction. So all of the components are correlated with wastewater and all of the components are correlated with nutrients (at least on RDA 1). What then makes you focus on a few components over others?**

*We have now normalized the PARAFAC components by DOC concentration to emphasize qualitative changes in DOM. This alleviates the problem of DOC concentration being the overriding determinant, which formerly resulted in the non-singular interpretation. In particular, component C4 on the one hand, and C6 and C7 on the other hand, now point in a different*

*direction than the remaining PARAFAC components. Furthermore, a new correlation analysis shows that C1, C3, C4 and C5 were not or negatively correlated with the TrOCs, whereas the other PARAFAC components were positively correlated with the TrOCs. Note that apart from their relation to the TrOCs, C2 and C7 (formerly C8) have been previously found in WWTP effluents.*

**Some more methodological details are needed on the PARAFAC process. Which software and how did you handle validation.**

*More details of the PARAFAC process were moved from the Supplement to the main body of the manuscript. This includes information on the software we used (L179) and how we handled validation (L181). Please also see the response further below.*

**An overlay of the split halves would be nice to see on the plot of the PARAFAC model. It helps the reader evaluate the quality of the model.**

*We have added this suggested figure to the appendix (Fig. B1). It shows results of the split-half validation of the final PARAFAC model.*

**A number of times DOM diversity is equated with functional diversity 'in the aquatic system' and I don't seem much evidence of this or a framework built for it. Connection DOM composition to ecosystem functioning is still pretty speculative (to be clear it is speculation I support, just still feel it has a long way to go). In particular the authors point to the diversity of DOM as an indicator of diversity or functional diversity in the aquatic system, but provide little evidence why it should be so. Clearly, DOM diversity is an indicator of the diversity of watershed processes both natural and anthropogenic – source diversity if you will. Is that function 'in the aquatic system' or function in the watershed? I would argue that it is the latter.**

*The rationale behind the linkage of DOM diversity and functional diversity is the idea that individual processes leave an imprint on DOM composition by generating or removing specific compounds. We agree with the reviewer that this is (still) speculative, so we have removed the concept of "functional diversity" from the revised manuscript (L374). The reviewer´s idea of DOM indicating catchment processes and not just within-system processes is now picked up in the form of aquatic-terrestrial coupling at several places in the manuscript.*

**However, with respect to the watershed you only ever look at a 50 M buffer (see detailed comments on this below).**

*Please see our response below.*

**In several places hydrology and runoff are presented as the cause of an observed relationship, but there is no mention of any aspect of the study design that evaluates hydrology. E.g. "…for example, was formerly connected to a sewage farm and appeared to be influenced by previously unrecognized storm water runoff that likely delivered inputs during heavy rain. "  No storm sampling was ever discussed, no pre-post sampling that would disambiguate this. There are just a lot of instances of statements and conclusions that are not or are not unambiguously supported by collected data.**

*The idea was to infer potential differences in runoff effects from site characteristics. To clarify matters, we have provided more site-specific information in the revised manuscript (susceptibility to wastewater inputs during storm events). This includes information on WWTP outlets added to figure 1, the main text (first in Methods) and Table A1. In addition, we have added a figure showing precipitation and flow data during the study (Fig. E1). Note that the presented hydrograph is rather typical of larger lowland rivers, which can react slowly to precipitation that could occur far away, yet it shows no major flood event during our study. The figure further shows that our campaigns were rather unaffected by major rain events during or just prior to our sampling periods, suggesting that smaller streams and ponds may not have been affected, either.*

**All that said. This is an interesting dataset and general question, I do encourage the authors to develop it further and focus on the clear and well-supported interpretations of the data.**

*Thank you very much.*

**Specific comments**

**39 'failure of citizens' … inappropriate and subjective statement. You blame the public, but have scientists properly communicated the issue to the public? Rather adversarial language that will only function to pit the general public against science. Why make an enemy?**

*We have rephrased the sentence to "This and the limited recognition of urban freshwaters as…" (L42).*

**47-48 Subjective. What is the purpose of monitoring? What is the endpoint. Often it is something much larger like ensuring healthy available habitat for human or animal use. If a primary driver of healthy habitat for animals is the availability of oxygen in the water, is that really a 'narrow focus' or is it the focus that is appropriate for monitoring given**

**the monitoring goals. I think you would be better arguing that high resolution approaches can expand the suite of bigger picture ecosystems states that can be monitored with DOM.**

*We have rephrased this section to "This focus is at odds with the extreme diversity of DOM observed in freshwaters, where thousands of compounds can be chemically distinguished (Kellerman et al., 2014; Peter et al., 2020; Stanley et al., 2012). This high diversity and the strong spatio-temporal variation of DOM composition suggest much potential for DOM characteristics to provide insights into the state of freshwater ecosystems in water quality assessment and monitoring. In fact, additional insights into freshwater ecosystems may be gained if the very high diversity of DOM can be used to inform about water quality for ecosystem assessment and monitoring purposes" (L50-55).*

**78-80 More info on this. What about these sites, what type of pollution do they represent. All the same type/intensity, different types?**

*We have added more information on the sites in Table A1 and Fig. 1a: nature of the sites (i.e. natural vs artificial), channelization, influence of WWTP effluents (L86-92).*

**84-86 Why only a 50 buffer? Why not a series of buffers to determine what the spatial scale is that is most relevant. The water interconnections in an urban ecosystem are complex, I doubt 50m captures the reality of the source areas. See Kaushal and Belt 2012.**

*The 50-m strips were supposed to capture influences in the vicinity of the sites (i.e. influences of the riparian zone and somewhat farther away) but not from the whole catchment, which is difficult to define in urban areas. This choice enabled us to distinguish between urban sites adjacent to paved surfaces and others in green spaces. Tufekcioglu (2020) and Johnson (2005) used buffer zones of similar size and a study on ponds by Declerck (2006) considered a range of widths (ranging from 50-3200 m) and found 50 and 100 m to be most appropriate to assess land-cover effects.*

**105-110 Did you collect and process any blanks?**

*We used ultra-pure water as a blank. This information was moved from the Supplement/Appendix to the main body of the manuscript (L140).*

**105-110 Was iron measured in any of these samples? This can have significant effects on optical DOM determination and is often elevated as it runs through urban infrastructure.**

*We added a short text passage and a figure on the potential interference of Fe with optical DOM signatures. (L151-159, Fig. E2).*

**123 A few things here. This is almost universally abbreviated FI and not FIX. You are using the wavelengths for you calculations for samples corrected for instrumental bias. This is appropriate. However, the citation you reference here was based on FI values calculated from a the old wavelengths that were not corrected for instrumental bias. McKnight updated this in Corey et al. 2010 and it makes a significant difference in the reference values of allochthonous and autochthonous endpoints. Lastly in heavily impacted urban systems, the classical interpretation of FI as developed by McKnight may simply not be applicable. You may be getting a 1.2 or a 1.9, but it may not mean the same thing as it would in a more natural system.**

*FIX has been replaced by FI (L168, Table B2) and Cory et al. (2010) (L171) is now used as the core reference.*

**126-127 Would like to see the split half validation overlaid on this PARAFAC model (Figure A1). Overall more details on the PARAFAC modelling process used would be nice.**

*We have added the split-half validation in the Appendix (Fig. B1), as suggested. The PARAFAC modeling details were moved from the Supplement to the main text and expanded particularly to provide information on cross-validation (L176-182). See also our response above.*

**286-288 Does it reflect high functional diversity across the 'aquatic network'? So far it would seem to suggest a variety of inputs or a diversity of input. I don't know if it says anything about what is going on in terms of fucntional/metabolic processes in the aquatic network. Also consider what 'functional diversity' means and what is 'desirable' vs. 'undesirable.' High functional diversity might be due to the wide range of degradation states that stream in an urban landscape may be experiencing.**

*Yes, both internal and catchment processes leave an imprint on DOM composition. Our point here is simply that high DOM diversity translates to potentially high information content about those processes. To avoid misunderstandings, we removed a part of the sentence (L350-351).*

**306-308 Could be, but you have provided no information on the hydrologic conditions at the time of sampling. Also within a season you haven't sampled during runoff conditions and during 'base flow' conditions to determine if there is a difference.**

*We have clarified that none of our samples were taken during intense precipitation or high-flow conditions (Fig. E1, L401-403) and that we refer to legacy effects due to site characteristics (e.g. impervious surface area). See also our response above.*

**313-315 weak inference. All of your components ordinate in the same direction of TrOCs. Also how did you establish the link to WWTPs. Is it just based on what other people said who found similar looking components?**

*We have strengthened the point by replacing the PCA based on TrOC data by an aggregate measure of TrOC abundance (L244) and by normalizing the PARAFAC data to express them as a proportion of total DOC (L249) and thus emphasize qualitative differences in DOM composition. We establish the link to WWTP by studying the correlation to the TrOC and also checking literature.*

**316-317 I would think that the greater abundance of light might be as big or a bigger factor than nutrients.**

*We have revised the paragraph to clarify our point that high nutrient levels were related to WWTP effluents, but did not drive autochthonous primary production as might be expected (L3411-412).*

**320-321 Again, not sure I see where that statement comes from. All of your DOM components ordinate in the same direction not just C2 and C8.  C1,2,4,5,7,6,8 (what happened to C3?) are all pretty well correlated with elevated nutrients on the primary RDA axis. It just seems like increased fluorescence is associated with increased nutrients.**

*The C3 label was inadvertently omitted in the graph but has now been included. The problem of the lack of differentiation among PARAFAC components has been alleviated by normalizing the PARAFAC data, as explained above.*

**340-341 why would you propose green space as a proxy for paved surfaces when you said you measured paved surfaces earlier?**

*In the revised manuscript, we discuss the role of green spaces in a more direct way and we removed the idea of using green space as a proxy of paved surfaces (337-340).*

**353 I don't know if your map is showing urban heterogeneity or not. I mean, none of this is clearly linked to urban influences (clearly some of it has to be). I just don't think the data and analyses you have presented lead to strong support for this statement.**

*Different colours in figures 1b and 1c mean different DOM compositions. The fact that these colours do not cluster indicates that the maps illustrate rather small-scale spatial turnover of DOM composition across the city. We have added this point in the legend of figure 1.*

**374-375 what do you mean by that? This study is based on single grab samples and average data? Most monitoring is part of a broader survey. This needs to be clarified.**

*We have clarified that we refer to temporal changes in DOM composition (i.e. compositional turnover) and the potential of using DOM composition as a complementary integrative measure for urban freshwater assessment and monitoring (L473-476).*

**384 How are you coming to this conclusion? You have presented no information that you ever sampled storm runoff?**

*This conclusion is based on very high levels of various variables, especially $NH_4^+$ concentrations, and historical information that the site used to receive stormwater runoff. This information is now included in the expanded site description (L89, Fig. 1, Table A1).*

**385 This is the first time it is mentioned. You should talk about this up in the sites section of the methods. Overall, a map showing the location of WWTPs would be very helpful. The WWTPs are being treated as a bit of an afterthought in the analysis when I feel like you should be framing your study and analysis around them.**

*As suggested, we have added information on potential WWTP influences to figure 1a and Table A1.*

**388-389 how do you know it "actually" received the inputs anything you have showing the hydrologic connectivity to a WWTP would be appreciated.**

*We have added information about the sites that could have received WWTP effluents upstream (L87-92, Fig. 1, Table A1). See also responses above.*

**404-405 What was actually detected? Which optical properties? Fluorescence? All the components ordinate in the same direction. TrOCs seem to be more a function of increasing DOC fluorescence overall. In this particular case for Berlin, I would then argue that the simplest thing to do is to measure FDOM fluorescence as an aggregate value as opposed to the finer resolution.**

*Simple optical data detected some PARAFAC components (C2 and C7) that were previously detected in WWTP effluents, and which were also correlated to the TrOCs. As mentioned*

*above, in the new PCA with normalized components, they are no longer pointing all in the same direction.*

*References*

*Cory, R. M., Miller, M. P., McKnight, D. M., Guerard, J. J., and Miller, P. L.: Effect of instrument-specific response on the analysis of fulvic acid fluorescence spectra, Limnology and Oceanography: Methods, 8, 67-78, https://doi.org/10.4319/lom.2010.8.67, 2010.*

*Declerck, S., De Bie, T., Ercken, D., Hampel, H., Schrijvers, S., Van Wichelen, J., Gillard, V., Mandiki, R., Losson, B., Bauwens, D., Keijers, S., Vyverman, W., Goddeeris, B., De meester, L., Brendonck, L., and Martens, K.: Ecological characteristics of small farmland ponds: Associations with land use practices at multiple spatial scales, Biological Conservation, 131, 523-532, https://doi.org/10.1016/j.biocon.2006.02.024, 2006..*

*Johnson, M.R., and Zelt, R.B., Protocols for Mapping and Characterizing Land Use/Land Cover in RiparianZones: U.S. Geological Survey Open-File Report 2005-1302, 22 p., 2005*

*Tufekcioglu, M., Schultz, R. C., Isenhart, T. M., Kovar, J. L., and Russell, J. R.: Riparian Land-Use, Stream Morphology and Streambank Erosion within Grazed Pastures in Southern Iowa, USA: A Catchment-Wide Perspective, Sustainability, 12, 6461, 2020.*

**ANSWER REVIEWER 2**

**General Comments**

**González-Quijano et al.'s manuscript seeks to understand how dissolved organic matter (DOM) pools are structured in an urban ecosystem. The manuscript further describes how well DOM quality relates to conventional water quality monitoring measurements and asks if DOM would be a useful indicator of water quality. The study used three different approaches to characterize the organic matter pool. I think the manuscript would be of interest to a broad audience and provides a useful and complex dataset. The results did well at describing the main multidimensional patterns in the data without focusing to heavily on individual data specifics. I think the multivariate approach used in the paper has merits, but I think adjustments to the approach would be useful.**

*Thank you.*

**First, the trace organic compounds (TrOC) PCA overcomplicated the manuscript and made it challenging to connection microcontaminant loads with urban pollution.**

*We replaced the PCA of the trace organic compounds (TrOCs) by the mean of all TrOCs after standardizing the values to ensure equal weighting. This aggregation is justified by the strongly positive correlations between all TrOCs.*

**Second, I think the DOM PCA could be focused by reducing the number of variables used and by presenting the parallel factor analysis (PARAFAC) results as percent of Fmax or relative to DOC.**

*We reduced the number of variables used in the PCA by removing the short-wavelength slope and also by reducing the PARAFAC components from 8 to 7. Importantly, we now express the PARAFAC results relative to DOC concentration, as mentioned above.*

**I think that both of these changes to the DOM PCA would allow the PCA and RDA to better highlight the data and connections between urban water quality markers. These adjustments will likely also meaningfully influence how DOM optical properties related to mass spectrometry results.**

*Thank you for the excellent suggestions.*

**Finally, I think the manuscript highlights an important topic, using DOM optical properties as a management tool. The current framing of the manuscript could be altered to better bring out this point. I think the "DOM as a monitoring tool" argument would be strengthened by adding broader statements explaining what makes DOM ideal for management, adding a hypothesis around the RDA between urban pollution drivers, and more fully explaining what basic knowledge is missing.**

*The Discussion is organized in three sections, the last of which is devoted to the potential of DOM optical properties (and DOM composition in general) to improve bioassessment and monitoring in the management of urban water bodies. We do not argue that DOM is an "ideal" indicator but that it is useful to complement existing approaches. Given that we are still in an explorative phase, we are also careful not to be too specific about hypotheses in this specific regard, although we have added some information and arguments to strengthen this point.*

**Below I provide more detail around these main points for the author's consideration as well as other specific suggestions**

**Specific Suggestions**

**Abstract & manuscript framing – I think the study would be better set up if "basic" was explained in more detail. I am curious to know what connections are missing and how or why this information is needed to better understand DOM composition in urban ecosystems. I think it would be useful if the abstract reconnects the expected high DOM diversity to "filling in the basics"**

*We replaced the vague term "basic" by more specific information in the abstract (L17-19). Causal relationships between processes (functions) in urban settings and their imprints on DOM composition and dynamics are still poorly known. Furthermore, we point to the potential usefulness of DOM composition for water quality monitoring.*

**Discussion section 4.3 & manuscript framing – this seems like the main objective of the study and is a valuable argument to be made and supported. I think some of this framing is lost in the methods and results. It would be useful for the reader if perhaps more direct statements of hypotheses were made that connect more traditional water quality measures to the potential use of DOM in water quality monitoring.**

*We have focused on the potential added value of DOM monitoring and the use of DOM composition to infer processes, rather than exploring correlations with established water quality measures.*

**Methods, 2.1 study sites – I think it would be useful to provide the size cutoff for streams vs rivers as was done for lakes and ponds.**

*Rivers and streams were classified according to a width cutoff of 5 m. This fact was added (L82).*

**Methods - Given the differences in habitat and the location of primary producers in Rivers, Streams, Ponds, and Lakes, I am not certain CHL should be used as a proxy for trophic state. Benthic algae are often abundant in urban streams. I think CHL should be removed from RDA because I don't think it's a comparable measure of eutrophication between lotic and lentic systems.**

*The rivers in Berlin are relatively deep and slow-flowing. In contrast, although streams are also typically slow-flowing, much of their autotrophic biomass could indeed be benthic. This limits the value of chl concentration in the water column as a proxy of eutrophication. However, in the absence of more suitable indicators and because high chl values do indicate high productivity (while the inverse is not necessarily true), we still prefer retaining the variable - along with ammonium, nitrate and total phosphorus concentrations.*

**Methods - I did not understand what the hypotheses were in RDA around DOM drivers. It was unclear what possible drivers were measured and then how they were applied to see changes in DOM. Perhaps more explanation is needed for readers like me with less experience using RDA and also to strengthen this analysis' connection to using DOM in urban monitoring.**

*We have expanded the RDA paragraph in Methods (L256-258) to clarify the rationale behind using the RDA (besides the PCA and the Procrustes test), the associated hypotheses, and the selected predictors.*

**Methods & Result - There are an impressive number of variables determined in this study. Many of which correlate or are a proxy for the same type of measure. For example, molecular weight is approximated through three optical indices and measured more directly with liquid size-exclusion chromatography. Given the complexity of the manuscript's dataset, it might be easier for the reader if only one variable that measures or estimates a DOM property was used. In my experience, especially given the inherent correlation between DOM characteristics, redundancy of multiple variables targeting the same DOM attribute are not needed. For example, I suggest only using S275-295 or SR as the optical indicator of DOM molecular weight. For comparison, its fine to keep all measures in the appendix but, for the main body of the paper and multivariate analysis, I think it would be easier for the reader to understand the results if only S275-295 or SR was used as the optical indictor of size. One final note: Short slope (S275-295) should be positive (see Fichot & Benner 2012 L&O 57(5):1453).**

*We have removed a strongly correlated variable (short-wavelength slope) as suggested and reduced the number of PARAFAC components. This changed the PCA biplot only slightly. Thank you also for pointing out the correct sign for short slope, although this variable has been removed in the revised manuscript.*

**Methods & Results – I am not certain the TrOC PCA is necessary. The purpose is to show micropollutant load. I worry about the below detection limit impact of ponds on the PCA loadings and scores. It seems like an easier and more staightforward metric that would also show micropollutant load is to sum all TrOCs and report a total TrOC concentration. This way, the reader does not need to remember three different PCAs and use a relative measure of load, when the sum of TrOCs would provide an understandable indicator of micropollutant load.**

*As stated above, we now standardized all TrOC data to zero mean and unit variance to compute a mean concentration for each site while ensuring equal weighting of the variables. Then we used this mean concentration as an aggregate indicator of pollution by TrOCs in the RDA analysis. This approach is similar to calculating the sum, as suggested, and is justified by the strong positive correlations between all TrOCs. We also show the correlations by presenting the original PCA results in the Appendix.*

**Methods & Results – I think the DOM PCA would structure better around DOM quality if PARAFAC components were set as percentages of Fmax or relative to DOC rather than RU. RU tends to follow concentration rather than clearly line up to quality measures and I think this is why all PARAFAC components point in the same direction in the DOM PCA. For example, DOC was significantly higher in streams. The DOM PCA water body type clusters follow fairly well this DOC concentration gradient, with all the PARAFAC components increasing in intensity toward places that had higher DOC. This would make quantity rather than quality the main driving force behind the PCA loadings. I agree that there is some separation of allochthonous to autochthonous sourced DOM along PC1 but the PARAFAC components did not follow the expected pattern based on quality. I am guessing that making the PARAFAC components relative, will cause them to line up much better across the source gradient. Using PARAFAC components as a percent of total Fmax or standardized to DOC (RU/DOC) might also provide better overlap between optic and FT-ICR-MS properties of DOM and help DOM optical estimates of similar compositional properties better align in the PCA space. The last paragraph of the results describes the FT-ICR-MS results clearly and in summarized way the read can understand. However, the PC comparisons don't always match with the optical multivariate space. For example, C6 & C8 are negative on PC1 and positive on PC2 suggesting there is more protein in that quadrant, but the comparison with FT-ICR-MS indicates that N containing compounds are positively related to PC1. Similar conflicts arise with humics. I think the reason for this is the PARAFAC components are being driven by quantity rather than quality. By making PARAFAC relative, then the resulting re-analyzed PCA might track more expectedly with FT-ICR-MS patterns.**

*This is an excellent suggestion that we have implemented. All PARAFAC components are now expressed relative to DOC concentration to emphasize the qualitative DOM differences among sites. In the new PCA, some of the PARAFAC components are now distinctly different from the others on the biplot, while the remaining variables ordinate almost like in the original PCA. After expressing PARAFAC relative to DOC, components C6 and C7 (former component*

*C8), which are protein-like components, align perfectly in the PCA with g10 (peptides) from the FT-ICR-MS.*

**Discussion, around line 320 - These are interesting ideas and useful points. I wonder if the lower B:A in streams reflects that the WWTP degrades DOM and the effluent is highly processed, while in lakes and ponds there is more new production of DOM resulting in a higher index score.**

*Thank you for the idea. We have added the point to the Discussion (L418-419).*

**Figure 1 – I like the style of this figure. I think it shows the sites and results well. For the caption, the reader needs more information. At this stage in the manuscript, it is unclear what variables are included in the PCA scores and what the patterns mean. It would be useful for the reader if a plan language interpretation of each PC axis was given so the reader could quickly understand the pattern**

*The figure caption was expanded as suggested.*

**Supplemental Information – I am not certain why the SI is needed given the large set of appendices. Why not move table S1 to the appendix and incorporate the DOM analysis information into the main methods. At least for, FT-ICR-MS the SI methods are very similar to the main methods.**

*The supplement is now omitted and information moved to the main text or Appendix.*

**Table A4 - Should the FIX for rivers be 1.68 rather than 0.68? 0.68 is well outside the normal range for FIX and could indicate an issue with how EEMs were processed, contamination or scanning error, or a coding error for the calculations**

*Corrected. Thank you for spotting the mistake. The correct value is indeed 1.68.*